# The gut microbiota in infants of obese mothers increases inflammation and susceptibility to NAFLD

Taylor K. Soderborg[1], Sarah E. Clark[2], Christopher E. Mulligan[1], Rachel C. Janssen[1], Lyndsey Babcock[1], Diana Ir[3], Bridget Young[4,13], Nancy Krebs[4], Dominick J. Lemas[1,14], Linda K. Johnson[5], Tiffany Weir[6], Laurel L. Lenz[2], Daniel N. Frank[3], Teri L. Hernandez[7,8], Kristine A. Kuhn[9], Angelo D'Alessandro [10], Linda A. Barbour[7,11], Karim C. El Kasmi[12] & Jacob E. Friedman[1,7,11]

Maternal obesity is associated with increased risk for offspring obesity and non-alcoholic fatty liver disease (NAFLD), but the causal drivers of this association are unclear. Early colonization of the infant gut by microbes plays a critical role in establishing immunity and metabolic function. Here, we compare germ-free mice colonized with stool microbes (MB) from 2-week-old infants born to obese (Inf-ObMB) or normal-weight (Inf-NWMB) mothers. Inf-ObMB-colonized mice demonstrate increased hepatic gene expression for endoplasmic reticulum stress and innate immunity together with histological signs of periportal inflammation, a histological pattern more commonly reported in pediatric cases of NAFLD. Inf-ObMB mice show increased intestinal permeability, reduced macrophage phagocytosis, and dampened cytokine production suggestive of impaired macrophage function. Furthermore, exposure to a Western-style diet in Inf-ObMB mice promotes excess weight gain and accelerates NAFLD. Overall, these results provide functional evidence supporting a causative role of maternal obesity-associated infant dysbiosis in childhood obesity and NAFLD.

[1] Department of Pediatrics, Section of Neonatology, University of Colorado Anschutz Medical Campus, Aurora 80045 CO, USA. [2] Department of Microbiology and Immunology, University of Colorado Anschutz Medical Campus, Aurora 80045 CO, USA. [3] Department of Medicine, Division of Infectious Disease, University of Colorado Anschutz Medical Campus, Aurora 80045 CO, USA. [4] Department of Pediatrics, Section of Nutrition, University of Colorado Anschutz Medical Campus, Aurora 80045 CO, USA. [5] Department of Pathology, University of Colorado Anschutz Medical Campus, Aurora 80045 CO, USA. [6] Department of Food Science and Human Nutrition, Colorado State University, Fort Collins 80523 CO, USA. [7] Department of Medicine, Division of Endocrinology, Metabolism & Diabetes, University of Colorado Anschutz Medical Campus, Aurora 80045 CO, USA. [8] College of Nursing, University of Colorado Anschutz Medical Campus, Aurora 80045 CO, USA. [9] Department of Medicine, Division of Rheumatology, University of Colorado Anschutz Medical Campus, Aurora 80045 CO, USA. [10] Department of Biochemistry and Molecular Genetics, University of Colorado Anschutz Medical Campus, Aurora 80045 CO, USA. [11] Department of Obstetrics and Gynecology, Division of Maternal Fetal Medicine, University of Colorado Anschutz Medical Campus, Aurora 80045 CO, USA. [12] Department of Pediatrics, Section of Gastroenterology, Hepatology and Nutrition, University of Colorado Anschutz Medical Campus, Aurora 80045 CO, USA. [13] Present address: Department of Pediatrics; Allergy and Immunology, University of Rochester School of Medicine and Dentistry, Rochester, NY 14642, USA. [14] Present address: Department of Health Outcomes and Biomedical Informatics, University of Florida, Gainesville, FL 32610, USA. Correspondence and requests for materials should be addressed to J.E.F. (email: jed.friedman@ucdenver.edu)

A large and growing body of evidence supports the concept of developmental programming through which the maternal environment affects fetal and infant development, thereby altering the risk profile for disease later in life[1,2]. Childhood obesity is a world-wide epidemic, with recent models predicting that the majority of today's children (57%) will be obese by age 35[3], with half of childhood obesity occurring by age 5[4]. This parallels the rise in maternal obesity, which approaches 40%[5], strongly supporting a role for early intrauterine/postnatal exposure to contribute to metabolic risk. Obesity also heightens the risk for non-alcoholic fatty liver disease (NAFLD), which affects at least 30% of obese children[6]. Furthermore, half of the children with NAFLD have already progressed to the more severe non-alcoholic steatohepatitis (NASH) at diagnosis[7], indicating a strong inflammatory component in pediatric NAFLD.

The pathophysiology of pediatric obesity and NAFLD has not been resolved, but compelling data in animals and humans show consistent links between exposure to maternal obesity and childhood obesity/NAFLD[2]. The severity of childhood NAFLD correlates with maternal obesity[8], birthweight[9], and shorter duration of breastfeeding[8,10], even after adjusting for childhood body mass index (BMI). Previously, we reported that 2-week-old infants born to obese (Ob) mothers with gestational diabetes had a 68% increase in intrahepatic fat compared with infants born to normal-weight (NW) mothers, which correlated with maternal pre-pregnancy BMI[11]. Similarly, in our non-human primate model of maternal diet-induced obesity, fetal steatosis and postnatal inflammatory hepatic infiltration, along with gut microbial dysbiosis, were present in juvenile offspring even after shifting to a healthy diet at weaning[12–14]. Likewise, in rodent models, maternal high-fat diets induce fetal liver steatosis that rapidly progresses to inflammation and fibrosis postnatally[15–18]. While these studies suggest that the risk factors for pediatric obesity/NAFLD begin in early-life, possibly in utero, very little is known about the early drivers of obesity and NAFLD risk in infants, and if microbiome disruption plays a causal role.

During early-life, the human gut microbiota undergoes a process of ecological succession, where a systematic turnover of species culminates in the establishment of a relatively stable complex community[19]. Disruption of this early process has been linked with increased risk of childhood inflammatory diseases and contributes to increased obesity risk and metabolic disease[20–22]. Elevated maternal pre-pregnancy BMI is associated with an altered infant microbiome at 2 days[23], 2 weeks[24], 1 month[25], 6 months[26], and 2 years of age[26]. While maternal obesity is associated with significant alterations in the infant gut microbiome, studies correlating these changes with future obesity or immune dysfunction are frequently confounded by multiple disruptors of the microbiota such as cesarean delivery, perinatal antibiotic use, or postnatal feeding patterns. This represents a significant barrier to our understanding of how early gut dysbiosis might increase disease risk in humans.

Here, we study infants born to NW and Ob mothers; these infants were born vaginally, exclusively breastfed, and were without exposure to antibiotics after delivery, as previously described[24]. Using germ-free (GF) mice, we investigate the hypothesis that early gut dysbiosis noted in 2-week-old infants born to Ob mothers causes metabolic and inflammatory changes characteristic of obesity and NAFLD. Our results demonstrate that gut microbes from babies born to Ob mothers are sufficient to induce changes in intestinal permeability and hepatic metabolism, including inflammation and a dysfunctional macrophage phenotype in GF mice that might be causal factors underlying increased transmission of obesity and NAFLD risk in children born to Ob mothers.

## Results

**Subject characteristics.** Information on subject characteristics of the mothers and infants utilized for this study is shown in Table 1. We utilized a subset of 15 stool samples from a larger cohort of 2-week-old infants born to NW ($n = 7$) or Ob ($n = 8$) mothers, based on pre-pregnancy BMI. Gestational weight gain in NW mothers was greater than Ob mothers ($P = 0.0005$); however, both were within accepted guidelines (NW = 11.5–16 kg, Ob = 5–9.0 kg)[27]. Ob mothers at 2-weeks postpartum had approximately double the percent body fat mass compared with NW mothers ($P = 0.0007$), measured by BOD POD. At 2-weeks postpartum, maternal adiponectin levels were 60% lower in Ob mothers ($P = 0.06$). In this small subset of relatively healthy Ob mothers, no differences in fasting glucose, insulin, or HOMA-IR were observed (Table 1). No significant differences were found in birthweight or newborn percent fat in this small sample subset. Per self-reported questionnaires, maternal diet at 2-weeks postpartum was consistent between groups on all measurements (Supplementary Table 1).

**Inf-ObMB increases SCFAs and intestinal permeability.** We pooled stool samples from 2–3 infants per maternal group to colonize GF mice and performed multiple rounds of experiments as illustrated in Fig. 1. At the phylum level, mice colonized with stool from 2-week-old infants born to Ob mothers (Inf-ObMB) showed relatively higher levels of Firmicutes and less Bacteroidetes (Fig. 2a) compared with mice colonized with stool from 2-week-old infants born to NW mothers (Inf-NWMB), similar to the larger human infant cohort from which the samples were taken[24]. The ratio of Bacteroidetes to Firmicutes was 34.2 in Inf-NWMB-colonized mice compared with 5.64 in the Inf-ObMB group ($P = 0.005$). The decreased Bacteroidetes to Firmicutes ratio in the microbiota from Inf-ObMB is consistent with most animal models of obesity[28,29] and some[30,31], but not all, human studies of established obesity. At the class level, Inf-ObMB mice had significantly less Gammaproteobacteria ($P = 0.01$; Fig. 2b), consistent with the infant stool composition, published previously[24]. In addition, Inf-ObMB mice had increased Clostridia ($P = 0.00001$; Fig. 2b), a class of Firmicutes, which has been previously associated with adolescent obesity[32] and NASH[33]. Colonization of GF mice with Inf-ObMB produced a significant

**Table 1 Characteristics of infants and their mothers at birth and 2-weeks postpartum**

|  | Normal-weight | Obese | P-value* |
|---|---|---|---|
| Maternal age (years) | 30.9 ± 1.16 | 30.94 ± 1.83 | 0.98 |
| Pre-pregnancy BMI (kg m$^{-2}$) | 20.85 ± 0.41 | 34.28 ± 2.51 | 0.002 |
| Gestational weight gain (kg) | 14.43 ± 0.94 | 5.786 ± 1.05 | 0.0005 |
| Fat mass maternal (%) | 26.38 ± 1.78 | 43.05 ± 1.92 | 0.0007 |
| Maternal fasting glucose (mg dL$^{-1}$) | 77 ± 4.40 | 82.4 ± 1.86 | 0.26 |
| Maternal fasting insulin (ng mL$^{-1}$) | 6.5 ± 0.5 | 10.6 ± 2.89 | 0.25 |
| Maternal HOMA-IR | 1.25 ± 0.16 | 2.18 ± 0.61 | 0.23 |
| Maternal adiponectin (µg mL$^{-1}$) | 11.68 ± 2.97 | 4.6 ± 0.67 | 0.06 |
| Infant sex (male/total) | 6/8 | 3/7 | N/A |
| Infant birthweight (kg) | 3.33 ± 0.21 | 3.37 ± 0.41 | 0.89 |
| Infant weight 2-weeks postpartum (kg) | 3.69 ± 0.31 | 3.61 ± 0.20 | 0.82 |
| Fat mass, Infant (%) | 11.85 ± 1.62 | 12.45 ± 1.62 | 0.80 |

Results are expressed as mean ± SEM
*P-value assessed by Student's t-test

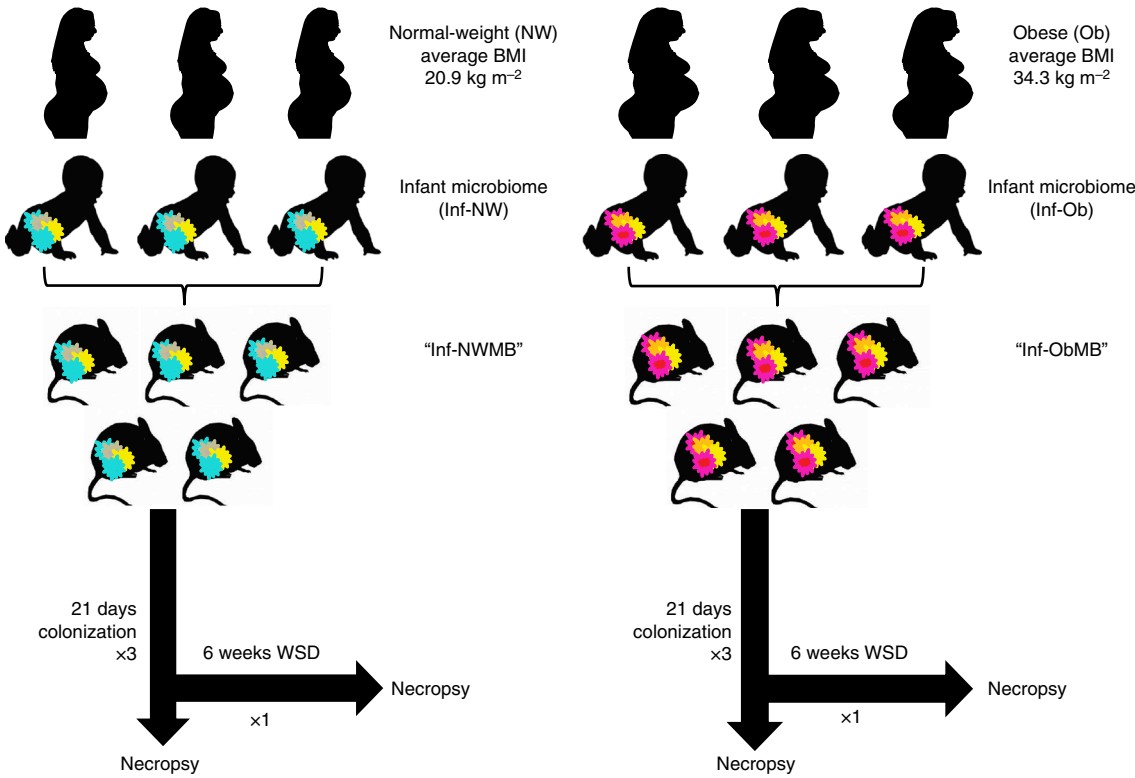

**Fig. 1** Diagrammatic representation of experimental design. Stool from three different 2-week-old infants born to normal-weight (NW) or obese (Ob) mothers was pooled in an anaerobic chamber. Pooled samples from each group, Inf-NWMB or Inf-ObMB, were gavaged into 3–5 GF mice per group and the inoculated mice were colonized for 21 days on a chow diet. This experiment was repeated three times using unique pooled samples of infant stools for each round of colonization. To study the effect of WSD, a fourth experiment was performed with 4 GF mice per group and following the 21-day colonization, mice were switched to a WSD for 6 weeks

increase in cecum short-chain fatty acids (SCFA) acetate, propionate, and butyrate concentrations compared with Inf-NWMB mice (1.6-, 1.8-, 5.7-fold change; $P = 0.0005$, $P = 0.03$, $P = 0.002$, respectively; Fig. 2c), similar to the fold changes in infants from which the samples were derived[24], in childhood obesity[34], and in adults with NAFLD[35].

The infant intestinal microbiota plays a key role in establishing gut barrier formation and permeability[36]. To determine whether colonization of GF mice with Inf-ObMB altered gut barrier gene expression, we measured the expression of genes encoding intestinal tight junction proteins, zona occludens 1 (*Tjp1*) and occludin (*Ocln*). *Tjp1* expression was significantly decreased ($P = 0.02$) in Inf-ObMB mice compared with Inf-NWMB mice (Fig. 2d), while *Ocln* trended lower ($P = 0.1$). To further investigate whether an increase in intestinal permeability accompanied the change in tight junction expression, intestinal permeability was measured in Inf-NWMB and Inf-ObMB mice by performing oral gavages with fluorescein isothiocyanate (FITC)-dextran and measuring translocation of fluorescence into the plasma. We found that intestinal permeability was significantly higher in Inf-ObMB mice ($P = 0.017$; Fig. 2e). We next examined whether elevated gut permeability was associated with bacterial translocation out of the gut. We quantified total bacteria in Inf-NWMB and Inf-ObMB mouse livers through the measurement of total 16S DNA, as 16S is only found in bacterial DNA. Significantly more 16S DNA was measured in Inf-ObMB livers compared with Inf-NWMB ($P < 0.0001$; Fig. 2f). In addition, we measured bacterial outgrowth from the livers of a subset of the colonized mice under one set of bacterial growth conditions and media. A trend for more bacterial growth was noted in livers from Inf-ObMB-colonized mice compared with Inf-NWMB mice ($P = 0.17$; Supplementary Fig. 1A), with greater

variation in four types of bacteria in Inf-ObMB livers compared with Inf-NWMB livers, based on differences in colony color, size, and morphology (Supplementary Fig. 1B).

**Inf-ObMB alters bile acid composition and metabolism**. Given the important role of gut microbiota in the metabolism of bile acids (BA)[37], we investigated the BA content in liver and feces from the colonized mice. No significant differences in liver BA content were noted between Inf-ObMB and Inf-NWMB mice (Fig. 3a, Supplementary Fig. 2A, B). Increased losses of primary ($P = 0.03$), secondary ($P = 0.03$), conjugated secondary ($P = 0.006$), and total ($P = 0.08$) BAs in feces were observed in Inf-ObMB mice compared with Inf-NWMB mice (Fig. 3b, Supplementary Fig. 2C, D). Hepatic mRNA expression of the rate limiting enzymes in BA synthesis, *Cyp8b1* ($P = 0.01$) and *Cyp7a1* ($P = 0.07$), were increased in Inf-ObMB (Fig. 3c). Moreover, hepatic BA exporter *Bsep* (*Abcb11*) expression was reduced ($P = 0.02$) and the hepatic BA importer *Ntcp* (*Slc10a1*) expression was elevated ($P = 0.05$) in Inf-ObMB mice (Fig. 3c). This fecal loss of BAs with upregulation of liver BA synthesis and shift in importer/exporter genes is consistent with reports in adult NAFLD[38]. The BA receptor *Fxr* (*Nr1h4*) gene expression was significantly elevated ($P = 0.03$) in livers of Inf-ObMB mice. However, neither downstream targets *Shp* (*Nr0b2*) nor *Hnf4a* differed between the groups (Fig. 3c), consistent with an altered FXR-SHP signaling axis, as reported in NAFLD and NASH[38].

**Inf-ObMB causes inflammation and impairs macrophage function**. Microbes in early-life play a key role in body weight gain and establishing proper immune function[39]. We found that mice colonized with Inf-ObMB showed a significant increase in

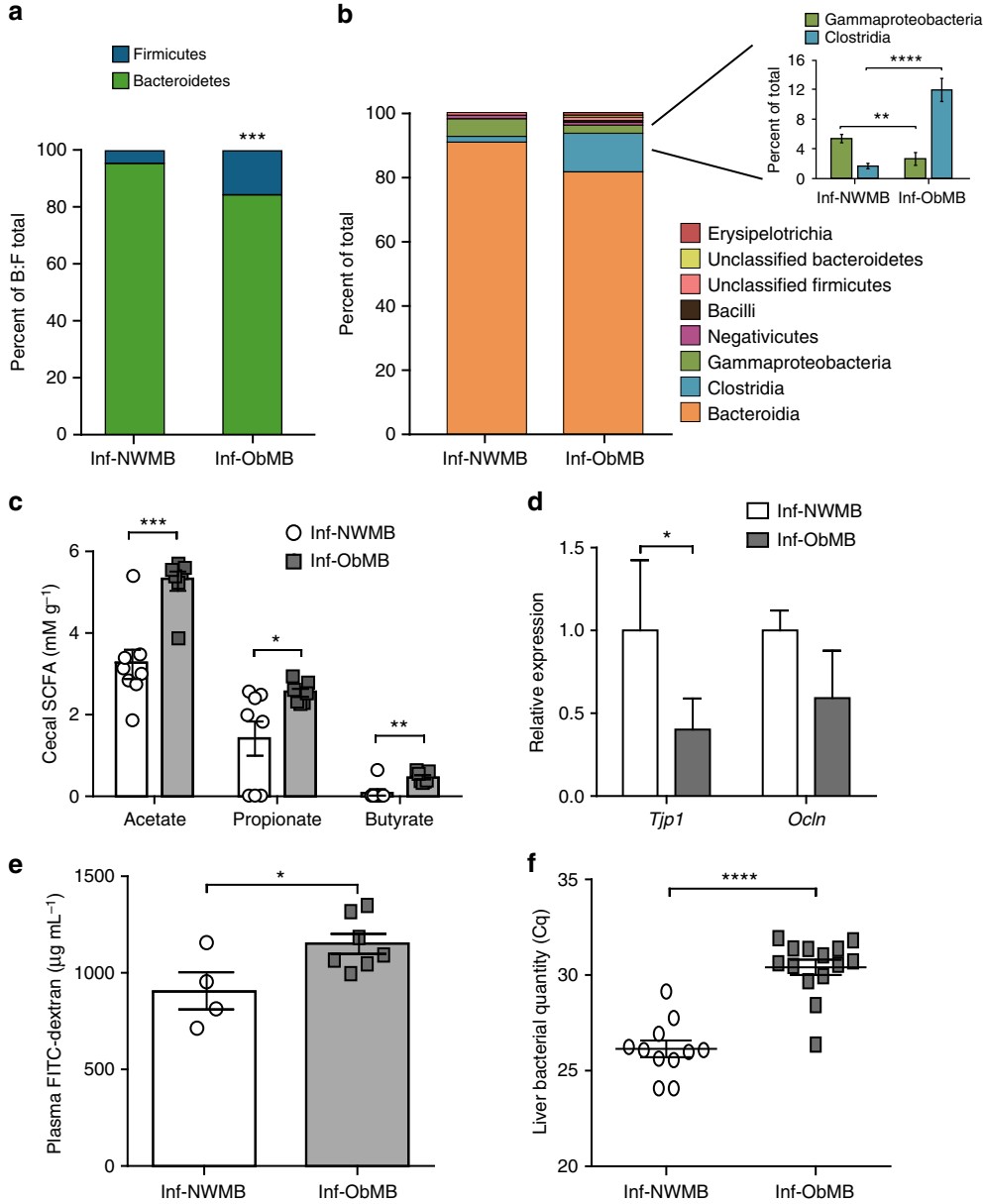

**Fig. 2** Inf-ObMB alters gut microbiome and increases intestinal permeability. Bacteroidetes to Firmicutes (B:F) ratio in the cecal microbiota depicted as percent of total of these two phyla (**a**) and class level relative abundance (**b**), with the percent of total highlighted for Gammaproteobacteria and Clostridia, after 21 days of colonization; $n = 6$ for both Inf-NWMB and Inf-ObMW groups. **c** Acetate, propionate, and butyrate SCFA concentrations measured in the cecum; $n = 8$ for Inf-NWMB, $n = 7$ for Inf-ObMB. **d** Relative mRNA expression of *Tjp1* and *Ocln* measured in the colon, normalized to *Gapdh* and *18S* rRNA; $n = 8$ for Inf-NWMB, $n = 7$ for Inf-ObMB. **e** FITC-dextran translocation into plasma; $n = 4$ for Inf-NWMB, $n = 7$ for Inf-ObMB. **f** Quantity of 16S DNA in liver; Cq, quantification cycle; $n = 11$ for Inf-NWMB and $n = 14$ for Inf-ObMB. **c**–**f** Data are presented as mean ± SEM. *$P < 0.05$, **$P < 0.01$, ***$P < 0.005$, and ****$P < 0.0001$ by Student's *t*-test

adipose tissue depot mass ($P = 0.0004$) without a change in overall body weight (Fig. 4a) or hepatic triglycerides (Fig. 4b) compared with Inf-NWMB mice. While liver triglycerides were not elevated, the hepatic lipid beta-oxidation gene *Cpt1a* was significantly increased ($P = 0.02$), along with a trend toward increased mRNA expression of the major lipid metabolism regulator *Ppara* ($P = 0.08$) and lipid storage gene *Pparg2* ($P = 0.08$) in the liver of Inf-ObMB mice (Fig. 4c). Regarding inflammation, we found that GF mice colonized with Inf-ObMB exhibited increased hepatic gene expression of *Xbp1s* ($P = 0.01$), a marker of endoplasmic reticulum stress, and increased *Tnf* cytokine expression ($P = 0.02$) compared with Inf-NWMB-colonized mice (Fig. 4c). In addition, expression of proinflammatory cytokines

*Ifnb1* and *Il6* mRNA was 25–50% greater in the livers of Inf-ObMB mice, although these did not reach significance (Fig. 4c).

Increased gut permeability and the proinflammatory state of the liver in Inf-ObMB mice prompted us to investigate macrophage phenotype and function. We examined the effects of Inf-ObMB colonization on liver macrophage content by flow cytometry. We identified resident macrophages (Kupffer cells) as F4/80 + hi/CD11blo, which were elevated in Inf-ObMB mice from 44 to 68% ($P = 0.06$), while recruited macrophages, identified as F4/80 + lo/CD11bhi, increased from <1 to 5.2% ($P = 0.2$). Overall, this resulted in a significant increase in the total macrophage pool in Inf-ObMB-colonized mice ($P = 0.01$; Fig. 4d). Altogether with increased mRNA for inflammatory cytokines and increased

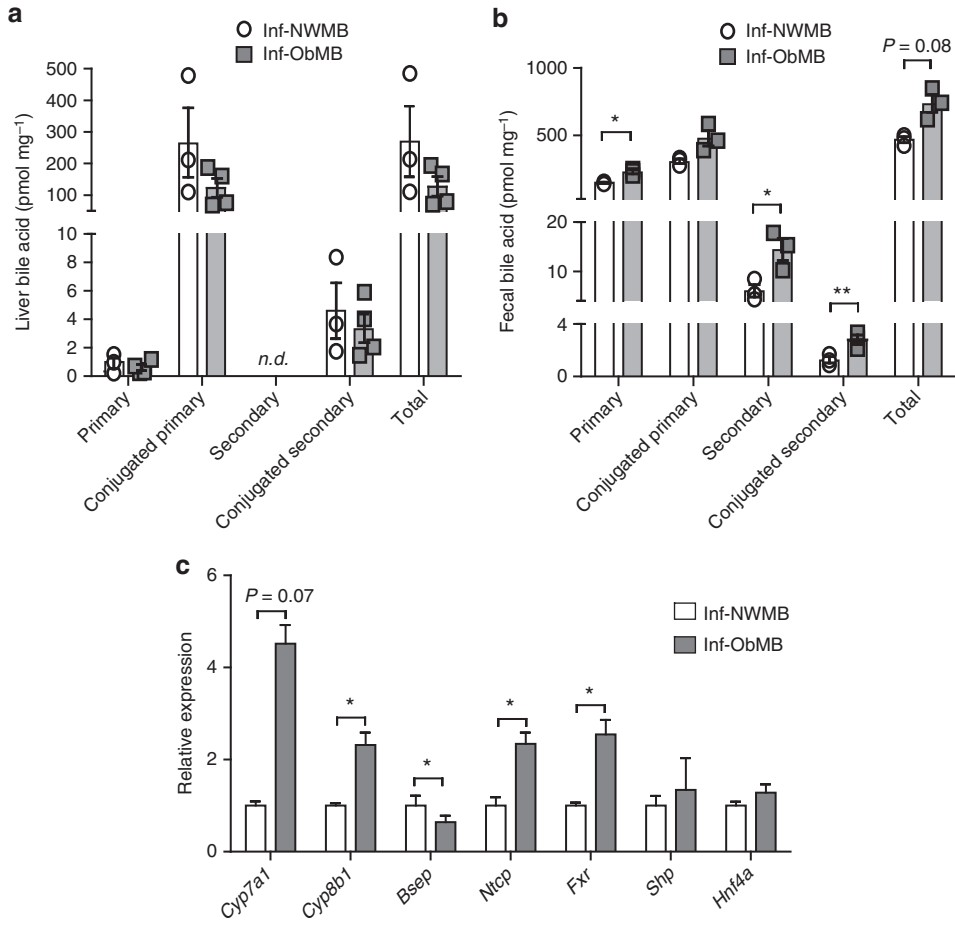

**Fig. 3** Inf-ObMB increases fecal BA loss and alters hepatic BA homeostasis genes. Concentration of primary, conjugated primary, secondary, conjugated secondary, and total BAs in the liver (**a**) and feces (**b**) in Inf-NWMB and Inf-ObMB mice; $n = 3$ for Inf-NWMB, $n = 4$ for Inf-ObMB for liver; $n = 3$ for both Inf-NWMB and Inf-ObMB for feces. **c** Hepatic gene expression of BA synthesis enzymes *Cyp7a1* and *Cyp8b1*, exporter *Bsep*, importer *Ntcp*, BA receptor *Fxr*, and its downstream targets *Shp* and *Hnf4a* in Inf-NWMB and Inf-ObMB mice, normalized to *Gapdh* and *18S* rRNA; $n = 6$ for Inf-NWMB, $n = 7$ for Inf-ObMB. Data are presented as mean ± SEM. *$P < 0.05$ and **$P < 0.01$ by two-tailed Student's *t*-test

numbers of resident macrophages, we found greater periportal leukocyte infiltration in the liver of Inf-ObMB-colonized mice compared with Inf-NWMB (Fig. 4e). Quantification of histology using a modified Pediatric NAFLD Histological Score (PNHS)[40] was significantly higher in livers from Inf-ObMB compared with Inf-NWMB mice ($P = 0.017$ by Mann–Whitney *U*-test; Fig. 4f), driven predominately by an increase in portal inflammation in the Inf-ObMB livers ($P = 0.017$ by Mann–Whitney *U*-test; Supplementary Fig. 3A). No micro- or macro-steatosis was present in either group, as expected given chow diet consumption and the short 21-day colonization period.

Bone marrow-derived macrophages (BMDMs) can provide insight into changes that are programmed prior to differentiation into monocytes and eventual recruited macrophages. We, therefore, examined gene expression in BMDMs from Inf-NWMB and Inf-ObMB mice. At baseline, expression of *Il1b, Il6, Tnf,* and *Il10* were not different between groups. To test the responsiveness of BMDMs to lipopolysaccharide (LPS), a typical proinflammatory stimulus derived from gut microbiota, we exposed BMDMs to low-dose LPS (10 ng mL$^{-1}$) for 20 h. mRNA expression of *Il1b, Il6, Tnf,* and *Il10* were significantly or trending lower ($P = 0.009$, $P = 0.08$, $P = 0.005$, $P = 0.02$, respectively) in the BMDMs from mice colonized with Inf-ObMB compared with Inf-NWMB (Fig. 5a). An essential step in the macrophage resolution of inflammation by pathogens is phagocytosis[41]. To investigate the potential for Inf-ObMB to alter macrophage phagocytosis,

BMDMs were incubated with GFP-expressing *Listeria*. BMDMs from Inf-ObMB-colonized mice were significantly less efficient in phagocytosing *Listeria* compared with BMDMs from Inf-NWMB mice ($P = 0.02$; Fig. 5b, c), consistent with reduced bacterial clearance and inflammation in Inf-ObMB mice.

**WSD exacerbates inflammation and adiposity in Inf-ObMB mice.** To assess whether bacteria from Inf-ObMB can cause acceleration of obesity and NAFLD, we examined the effect of short-term Western-style diet (WSD) exposure in mice colonized with Inf-NWMB or Inf-ObMB. After just 6 weeks of WSD exposure, Inf-ObMB mice showed increased body weight ($P = 0.06$) and significantly increased fat mass ($P = 0.03$) and liver triglycerides ($P = 0.03$) compared with Inf-NWMB mice (Fig. 6a–c). Total macrophage numbers in the livers of mice on WSD were similar between the Inf-NWMB and Inf-ObMB groups (Fig. 6d). However, gene expression of proinflammatory cytokines *Il1b* and *Tnf* were significantly elevated (both $P = 0.02$) in hepatic recruited macrophages from Inf-ObMB mice, and importantly, mRNA expression of anti-inflammatory *Il10* was significantly decreased ($P = 0.02$) in these macrophages (Fig. 6e). Following 6 weeks of WSD, histologically, livers from Inf-ObMB mice demonstrated a mixed pattern of periportal and lobular inflammation and steatosis (Fig. 6f), which resulted in a near doubling of the modified PNHS compared with Inf-NWMB WSD

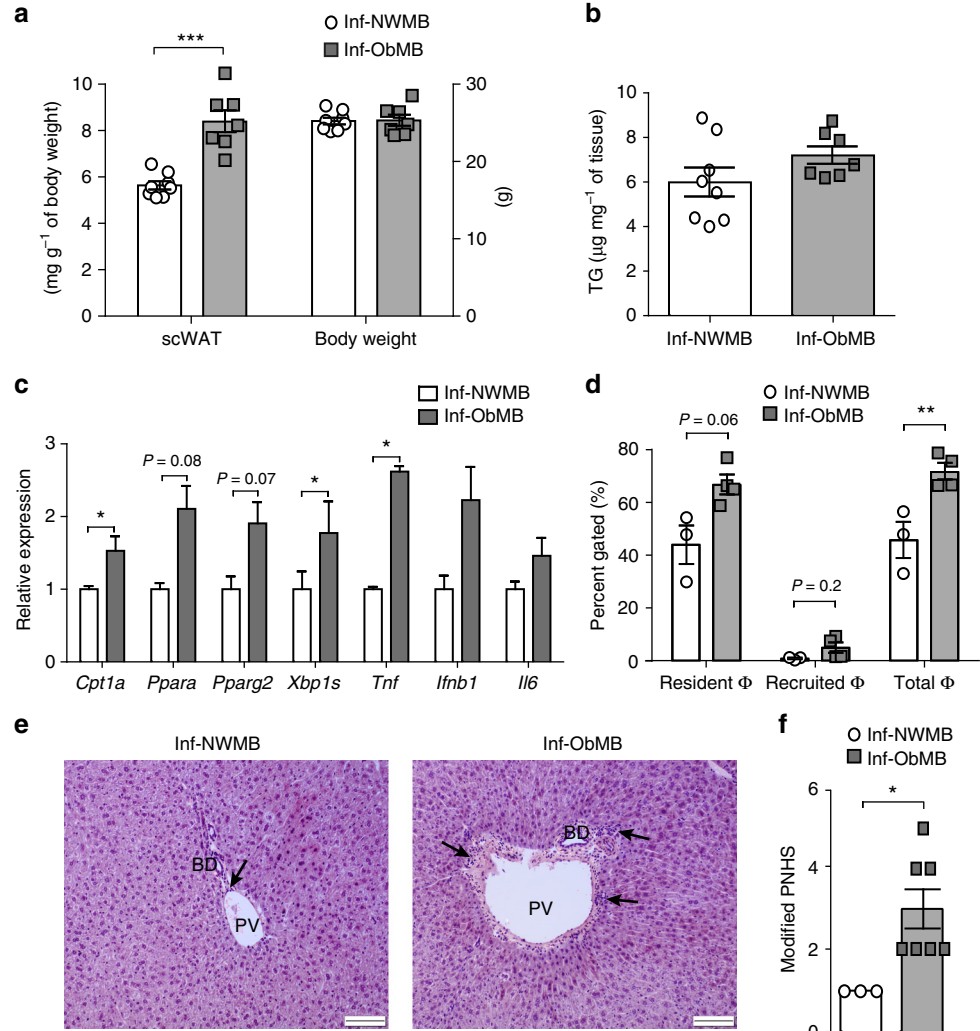

**Fig. 4** Inf-ObMB increases subcutaneous fat and liver inflammation. Weight of subcutaneous white adipose tissue (scWAT) and body weight (**a**), and liver triglycerides (TG) (**b**); $n = 8$ for Inf-NWMB, $n = 7$ for Inf-ObMB. **c** Hepatic expression of lipid regulatory and inflammatory genes normalized to *Gapdh* and *18S* rRNA; $n = 6$ for Inf-NWMB, $n = 7$ for Inf-ObMB (except *Tnf*, $n = 3$/group). **d** Hepatic resident, recruited, and total macrophage (Φ) quantities; $n = 3$ for Inf-NWMB, $n = 4$ for Inf-ObMB. **a–d** Data are presented as mean ± SEM. *$P < 0.05$, **$P < 0.01$ and ***$P < 0.005$ by two-tailed Student's *t*-test.
**e** Representative photomicrographs of H&E staining in the liver from Inf-NWMB and Inf-ObMB mice; black bars represent 100 μm, black arrows indicate infiltrating lymphocytes; PV portal vein, BD bile duct. **f** Modified Pediatric NAFLD Histological Score (PNHS) for Inf-NWMB and Inf-ObMB mice. $n = 3$ for Inf-NWMB and $n = 7$ for Inf-ObMB. Data are presented as mean ± SEM. *$P < 0.05$ by two-tailed Mann–Whitney *U*-test

mice ($P = 0.09$ by Mann–Whitney *U*-test; Fig. 6g; Supplementary Fig. 3B), consistent with increased hepatic cytokine expression results (Fig. 6e) and triglyceride levels (Fig. 6c).

Importantly, after 6 weeks of WSD exposure, the gut microbiome composition was no longer different between the Inf-NWMB and Inf-ObMB mice. At the phylum level, the Bacteroidetes to Firmicutes ratio was 7.71 in cecum from Inf-NWMB mice and 3.15 in Inf-ObMB mice (Fig. 7a). At the class level, the relative abundances of Gammaproteobacteria and Clostridia were no longer different between the Inf-NWMB and Inf-ObMB mice (Fig. 7b). In addition, cecal SCFA metabolites acetate, propionate, and butyrate were no longer different between the Inf-NWMB and Inf-ObMB groups (Fig. 7c).

## Discussion

The main finding of this study is that the altered gut microbiota in 2-week-old infants born to Ob mothers induced metabolic and inflammatory changes in the liver and bone marrow cells of GF

mice colonized with these dysbiotic microbes. Importantly, these mice were predisposed to accelerated weight gain and development of fatty liver following exposure to a WSD. Clinical data support correlations between pediatric obesity, NAFLD, and gut dysbiosis[42,43]; however, this is the first experimental evidence in support of the hypothesis that changes in the gut microbiome in infants born to obese mothers directly initiate these disease pathways.

Mice colonized with Inf-ObMB had increased hepatic endoplasmic reticulum stress as measured by *Xbp1s* gene expression, hepatic inflammation, and liver macrophage accumulation, consistent with the concept that Inf-ObMB provokes an inflammatory microenvironment in the livers of these mice. Inf-ObMB-colonized mice showed histological evidence for increased periportal inflammation, even on a chow diet. This periportal vein inflammation is seen in rodents with bacterial translocation[44] and in humans with advanced forms of pediatric NAFLD[40,45–47] and features of the metabolic syndrome[48], suggesting that it has clinical relevance as an early manifestation of leaky gut. We also

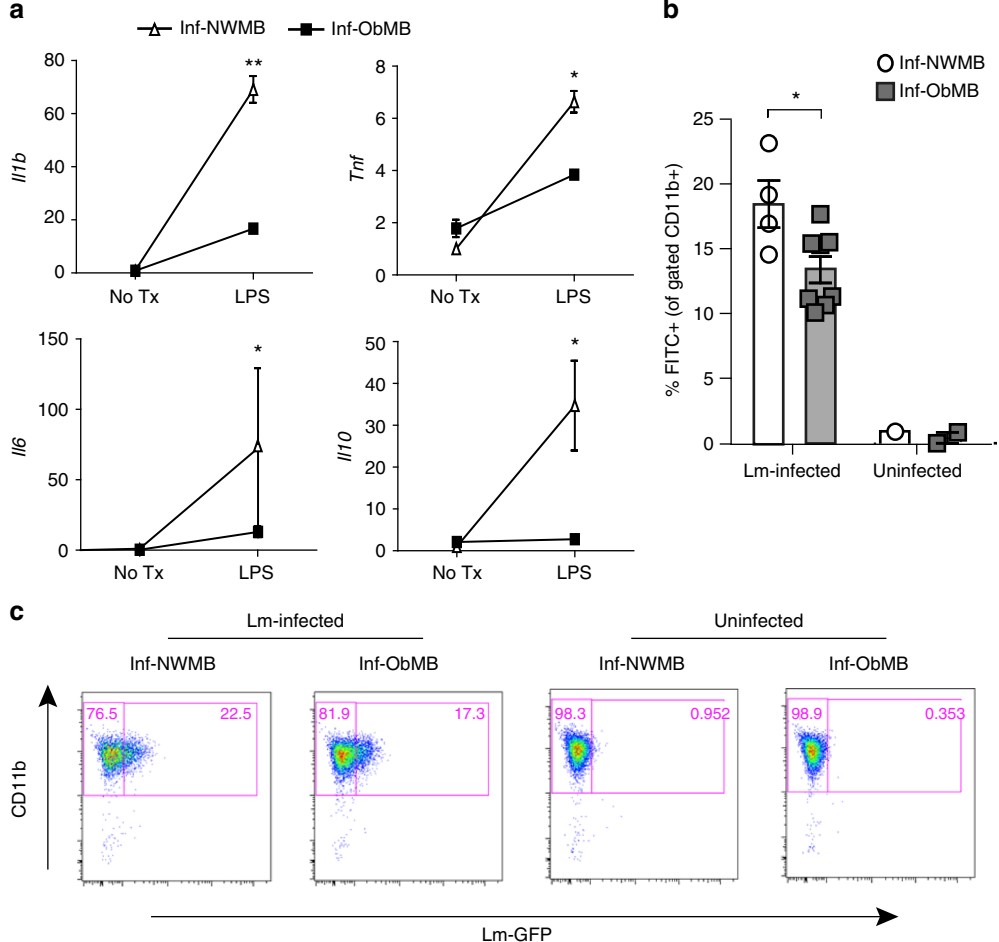

**Fig. 5** BMDMs from Inf-ObMB mice have impaired function. **a** Cytokine relative gene expression in BMDMs with and without 20-h low-dose LPS treatment depicted as relative to the Inf-NWMB no treatment (No Tx) group, with endogenous normalization to *Gapdh*; $n = 2$ for both Inf-NWMB and Inf-ObMB. CD11b$^+$/F4/80$^+$ BMDMs for GFP-expressing *Listeria* (Lm) uptake (**b**) with representative flow diagrams (**c**); $n = 4$ for Inf-NWMB, $n = 8$ for Inf-ObMB for Lm-infected; $n = 1$ for Inf-NWMB, $n = 2$ for Inf-ObMB for uninfected control. Data are presented as mean ± SEM. *$P < 0.05$ and **$P < 0.01$ by two-tailed Student's *t*-test

found evidence for reduced gut barrier gene expression and increased intestinal permeability as has been reported in children with established NAFLD[49,50].

In addition to hepatic inflammation, Inf-ObMB-colonized mice demonstrated a dysfunctional BMDM phenotype. The macrophages isolated from Inf-ObMB mice were hyporesponsive to LPS and had reduced ability to phagocytize *Listeria*, features consistent with an impaired anti-inflammatory, reparative macrophage phenotype[51]. Of interest, the reduced BMDM function was maintained even after differentiation in vitro for 7 days, suggesting that the dysbiotic Inf-ObMB gives rise to altered development of myeloid cell precursors. Myeloid cells are derived from precursors within the bone marrow and populate nearly all tissues, where they play crucial roles in the initiation of innate immunity, the resolution of inflammation, and tissue repair. In humanized gnotobiotic mice, colonizing microbiota and bacterial-derived metabolites, most notably SCFAs acetate, propionate, and butyrate[34,52,53], are known to affect myeloid cell progenitors of bone marrow origin[54]. The Inf-ObMB mice showed a significant elevation in all three measured SCFAs in the gut, similar to changes noted in the infant donor stool[24]. Butyrate induces a hypo-responsive inflammatory state in macrophages through its action as an HDAC inhibitor[52,55] and can also interfere with macrophage differentiation, maturation, and the ability of macrophages to prime other arms of the innate

and adaptive immune systems[56]. Although we did not measure butyrate in circulation, based on our data showing increased butyrate in the intestine, reduction in gut barrier integrity, and BMDM desensitization, we hypothesize that increased butyrate production by the infant gut dampens BMDM responsiveness to LPS[52,55].

Multiple studies have shown a broad effect of the gut microbiota on bone marrow hematopoiesis[54,57,58]. Kamimae-Lanning et al.[59] found that maternal obesity or high-fat diet in a murine model restricted the expansion and renewal of fetal hematopoietic stem cells by changing the transcriptional output of genes regulating metabolism, stress response, and proliferation. Importantly, these findings suggest that early microbes or maternal diet alter immune function through alterations in hematopoietic cell development and non-resolving, reparative macrophages derived from the bone marrow as well as the liver. Our findings of a dysfunctional macrophage phenotype are consistent with results recently described in cord blood-derived monocytes from infants born to obese mothers[60], due to epigenetic reprogramming. Here, we have identified a role of macrophages in the mechanism by which dysbiosis contributes to development of obesity and NAFLD; however, further evaluation of the recruited and resident liver macrophage populations will provide insight as to how these individual cell populations are altered by dysbiosis and highlight the mechanisms by which they contribute to NAFLD.

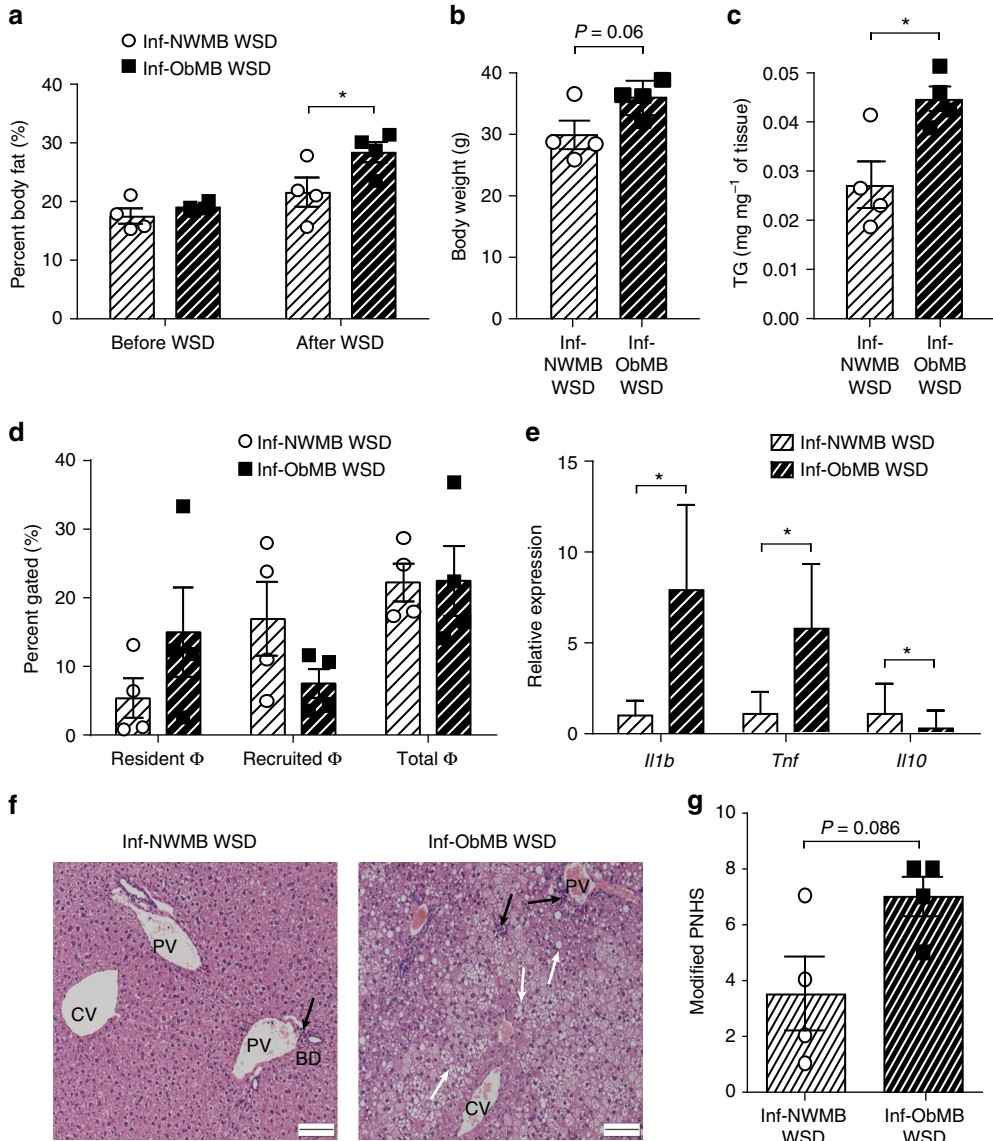

**Fig. 6** Inf-ObMB phenotype is exacerbated by WSD. **a** Percent body fat before and after 6 weeks of WSD. Inf-NWMB and Inf-ObMB mice after 6 weeks on WSD for body weight (**b**), liver triglycerides (TG) (**c**), liver macrophage (Φ) quantity (**d**), and proinflammatory *Il1b*, *Tnf*, and anti-inflammatory *Il10* gene expression in recruited macrophages from liver, normalized to *Gapdh* and *18S* rRNA (**e**). **a**–**e** n = 4 for both Inf-NWMB and Inf-ObMB. Data are presented as mean ± SEM. *P < 0.05 by two-tailed Student's *t*-test. **f** Representative photomicrographs of liver sections with H&E staining; black bars represent 100 μm, black arrows indicate infiltrating lymphocytes, white arrows indicate lipid droplets; CV central vein; PV portal vein; BD bile duct. **g** Modified Pediatric NAFLD Histological Score (PNHS) for Inf-NWMB and Inf-ObMB mice fed a WSD. n = 4 for both Inf-NWMB and Inf-ObMB. Data are presented as mean ± SEM. P = 0.086 by two-tailed Mann–Whitney *U*-test

Furthermore, the microbiome plays a key role in educating the multiple immune cell types, including Natural Killer T cells, T regulatory cells, as well as cells of the adaptive immune system[61]. Although these cell types were not investigated in this study, they are likely to also be altered by early dysbiosis.

Interestingly, we found a significant increase in primary and secondary BA levels in feces from Inf-ObMB mice, along with a striking increase in hepatic *Fxr*, *Cyp7a1*, and *Cyp8b1* mRNA expression (the rate limiting enzymes for BA synthesis) along with reduced *Bsep* (canalicular BA exporter) and increased *Ntcp* (basolateral BA importer) gene expression. These results suggest that colonization with Inf-ObMB leads to an increased loss of BAs in the feces, indicating reduced recirculation of BAs with partial hepatic compensation by increased synthesis through *Cyp7a1* and *Cyp8b1* expression and decreased excretion through *Bsep* suppression. Loss of BAs in the stool and reduced serum

BAs have been reported in adult[62] and early pediatric[63] NAFLD, respectively. The mechanisms by which Inf-ObMB dysregulates BA metabolism in the gut and liver, and the potential downstream effects on host physiology, merit further study.

The significant weight gain and hepatic steatosis in Inf-ObMB mice following a WSD suggest that Inf-ObMB primed the mice for obesity and NAFLD. Notably, the gut microbes and cecal SCFA levels after 6 weeks of WSD feeding were similar in the Inf-NWMB and Inf-ObMB mice. Despite a lack of persistent differences in the microbiome, the livers of WSD-fed, Inf-ObMB mice had recruited macrophages with a heightened proinflammatory cytokine response in the absence of increased macrophage numbers. This is consistent with programming towards a more proinflammatory and less reparative phenotype by the earlier microbial composition[64,65]. Although the mechanisms are poorly understood, animal models have suggested that disruption of the

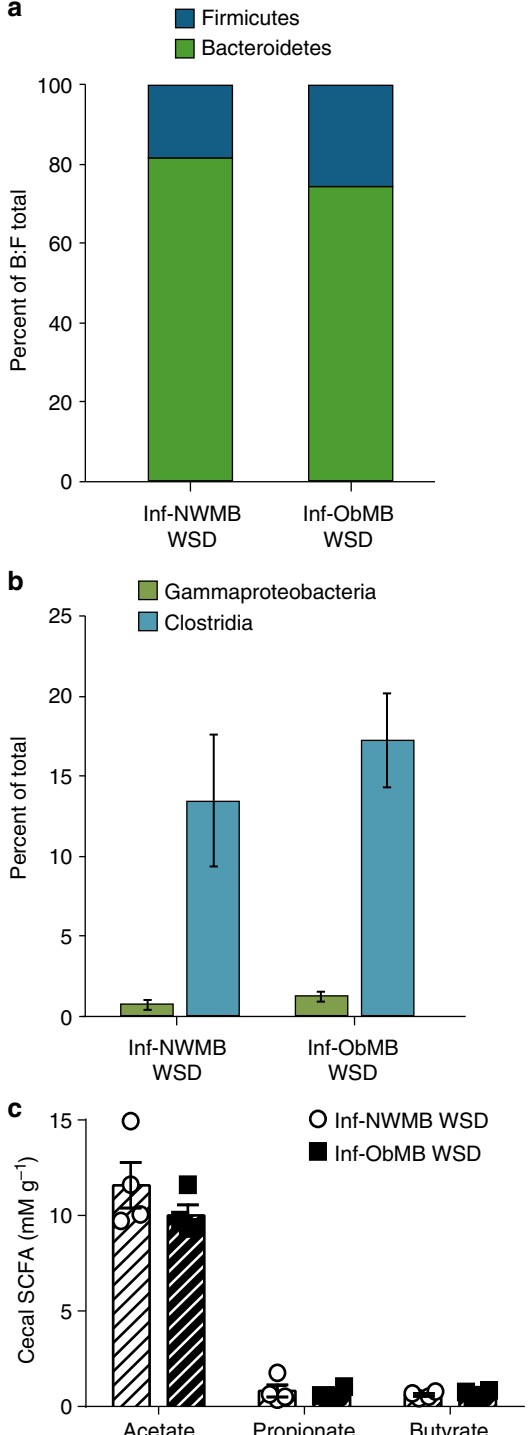

**Fig. 7** Inf-NWMB and Inf-ObMB gut microbiota are not different after 6 weeks on WSD. Cecal microbiota Bacteroidetes to Firmicutes (B:F) ratio (**a**), shown as percent of total of these two phyla, and Gammaproteobacteria and Clostridia relative abundances (**b**). **c** Acetate, propionate, and butyrate SCFA concentrations in the cecum in colonized mice fed a WSD; data are presented as mean ± SEM. **a–c** *n* = 4 for both Inf-NWMB and Inf-ObMB

microbiota in early-life profoundly alters the development of innate and adaptive immunity[61].

Other potential molecular pathways and factors by which dysbiosis mediates inflammation and progression to obesity and NAFLD include the production of bioactive lipids[66] and/or a shift

in gut microbial metabolic pathways[43]. In addition, microbial-associated or pathogen-associated molecular patterns (MAMPs or PAMPs, respectively) have been specifically shown to modulate hematopoiesis in GF mice[54]. Rodent studies have demonstrated a potential early causative role for an impaired intestinal immune system in the development of metabolic disease[67] and, therefore, intestinal macrophages warrant further consideration as a mechanistic pathway in this model. Our results indicate that macrophage dysfunction, in addition to other consequences of dysbiosis on liver inflammation, accelerate steatosis, and weight gain when exposed to WSD. Continued consumption of WSD beyond 6 weeks might enhance liver inflammation, injury, and more importantly, fibrosis in Inf-ObMB mice compared with Inf-NWMB mice, which were relatively protected from adverse effects of the WSD.

Our results emphasize that differences in gut microbes in infants of Ob mothers lead to a persistently compromised innate immune cell function when colonized in GF mice. We hypothesize that a lack of colonization by critical microbes in early-life interferes with proper myeloid cell development. In GF mice, colonization with *Escherichia coli* alone is sufficient to correct the defect in myelopoiesis observed in embryonic and bone marrow-derived myeloid progenitors, pointing to the critical role of early infant Gram-negative bacteria in regulating proper training of the innate immune system[54]. The reduction in Gammaproteobacteria (a known LPS producer) and increase in Clostridia (a known butyrate producer) noted in the infant donor stool[24] and in Inf-ObMB mice suggest that important pioneering bacteria necessary for proper immune development and function are compromised in infants born to Ob mothers. These studies suggest that decreased Gammaproteobacteria and/or increased Clostridia are important biomarkers in the stool of infants born to obese mothers that carry metabolic risk.

Overall, these results demonstrate that changes in the early gut microbiome composition in infants from Ob mothers lead to increased gut permeability, reduced macrophage phagocytic activity, and bacterial translocation to the liver that result in increased hepatic inflammatory responses and trigger NAFLD and excess weight gain in humanized GF mice. Our study design of pooling infant stool samples allowed us to create one inoculum for each round of colonization treatment group. Although it might have limited the variability seen between individual infants at this stage of development, the microbiota compositions of the Inf-NWMB and Inf-ObMB mice at 21-days post-gavage were significantly different and consistent with the major compositional differences in the NW and Ob infant donor stool reported previously[24]. This, along with our replication of the findings with three rounds of colonization using unique pools of stool for each round, strengthens the likelihood that our results are relevant for a larger human infant population.

We recognize the difficulty of extrapolating early macrophage programming and liver development to infants using humanized GF mice. However, our findings suggest that bacterial changes in infants of Ob mothers are associated with pathological changes that are permissive for obesity and accelerated risk for NAFLD. Further study of the relationships between dysbiosis, altered liver and bone marrow-derived macrophage populations, SCFAs, BAs, and NAFLD is likely to tease out these complexities and mechanisms by which the early microbiome contributes to metabolic disease risk. Limited evidence to date indicates that the maternal diet can influence both adiposity in early-life and postnatal immunity by altering the offspring gut microbiome[68], supporting a potentially modifiable risk factor and interventional strategy. This model can be used in future studies to identify important biomarkers in infants that can produce potentially disease-promoting or disease-protecting NAFLD phenotypes.

Future studies utilizing pre/probiotics, as well as understanding their bioactive metabolites that might prevent metabolic perturbations, are needed to modify the epidemic of childhood obesity and NAFLD risk in infants born to obese mothers.

## Methods

**Ethics statement**. All aspects of the clinical studies were approved by the Colorado Multiple Institutional Review Board (COMIRB) and the studies were registered at clinicaltrials.gov (NCT01693406 and NCT00826904). All animal studies were approved by the University of Colorado Institutional Animal Care and Use Committee (protocol number 00309) and carried out in strict accordance with the guidelines set forth by the Guide for the Care and Use of Laboratory Animals by the National Research Council.

**Human participants**. This study included subjects enrolled in ongoing longitudinal studies of mother-infant dyads at the University of Colorado Hospital (Aurora, CO) beginning in 2012. Informed consent for collection and study of infant stool was obtained from mothers. All mothers enrolled in this study were aged 25–35 years old, delivered vaginally, had normal glucose tolerance and had gestational weight gain within recommended IOM guidelines[27], and were not exposed to antibiotics throughout gestation and after delivery, with the following exceptions. One NW mother received antibiotic treatment during delivery (200 mL of 5 million units Penicillin G potassium in normal saline approximately 3 h prior to delivery). One mother in the obese group received a single dose of 1000 mg cefazulin IV in the immediate postpartum period. Offspring were born of a singleton pregnancy at term ($> 37$ weeks gestation) to mothers classified as NW ($< 25$ kg m$^{-2}$) or overweight/Ob ($> 28$ kg m$^{-2}$) based on pre-pregnancy self-reported BMI, and exclusively breastfed for the first 4 months of life. At 2-weeks postpartum, mothers' and infants' percent body fat were measured by air-displacement plethysmography (BOD POD and PEA POD, respectively; Cosmed, Rome, Italy). Mothers' blood was collected after an 8-h fast and plasma was isolated and frozen at $-80$ °C until analysis by the University of Colorado Hospital Clinical & Translational Research Center (CTRC) for measurements of glucose and insulin, from which HOMA-IR was calculated, and adiponectin. Maternal diet was quantified by self-reported 3-day diet records within 1 week of the 2-week postpartum visit. Records were analyzed by nutritionists in the CTRC Nutrition Services core using NDSR software (2016 version, University of Minnesota).

**Microbiome composition**. At the 2-week postpartum visit, mothers brought an infant stool sample, collected within 24 h of visit and saved in a frozen diaper at home (~ $-20$ °C), which was then aliquoted and stored at $-80$ °C. Cecal content was collected from colonized mice 21 days post-inoculation or following 6 weeks of WSD exposure and stored at $-80$ °C. Microbial DNA from all stool and cecum samples was extracted using the PowerFecal DNA Isolation kit (Qiagen). DNA was subjected to broad-range amplification using PCR of the V1V2 variable region in the 16S ribosomal RNA genes. Paired-end sequencing was performed using the Illumina MiSeq platform (version 2.3.0.8). Taxonomic classification was determined by aligning and classifying sequences with SINA classifier using the Silva 111 non-redundant database as a reference. Identical 16S rRNA sequences were grouped into operational taxonomic units (OTUs). Data were visualized using Explicet[69].

**Germ-free mice and colonization**. Figure 1 provides an overview of the experimental design. C57BL/6J male mice age 8–10 weeks were maintained in flexible plastic gnotobiotic isolators in the University of Colorado Anschutz Gnotobiotic Facility under a strict 12-h light–dark cycle and fed an autoclaved standard chow (16% kcal as fat; 2020SX; Envigo, Indianapolis, IN) ad libitum. GF isolators were routinely tested for sterility by culturing and pan-bacterial 16S rRNA gene PCR analysis of feces. Samples from 2-week-old infants born to 7 NW and 8 Ob mothers were used in these experiments. Colonization was performed using pooled infant stool samples from 2–3 infants from each group (Inf-NW and Inf-Ob) by diluting human stool samples resuspended in sterile, reduced PBS (6.7% wt vol$^{-1}$) in an anaerobic coy chamber. Each GF mouse was orally gavaged with 200 μL of sample under sterile conditions. Three groups of male mice were colonized with stool from either Inf-NW or Inf-Ob group. Mice were singly housed in the GF facility for 21 days ($n = 12$ per group). In a separate experiment, mice were colonized for 21 days then moved from the GF facility to the animal resource center and fed WSD (42% kcal as fat, 0.2% cholesterol, 34% sucrose; TD.88137; Envigo) ad libitum for 6 weeks ($n = 4$ per group). Body composition was determined using quantitative magnetic resonance (Echo MRI Whole Body Composition Analyzer; Echo Medical Systems, Houston, TX) 24–48 h prior to sacrifice. After 21 days of colonization or after 6 weeks on WSD, mice were fasted for 4 h, anesthetized using isoflurane inhalation, blood was sampled from the portal vein, and tissues (cecum, liver, subcutaneous adipose, colon, and hind-limb bone) were harvested, prior to euthanizing by cervical dislocation. All tissues were immediately frozen in liquid nitrogen and stored at $-80$ °C until further processed. After 21 days of colonization, mice were water-starved overnight and orally gavaged with FITC-dextran (100 mg mL$^{-1}$; 4 kDa; 44 mg 100 g$^{-1}$ body weight) dissolved in PBS, and plasma

was collected after 4 h. Using two aliquots of plasma in duplicate, FITC fluorescence (excitation 488 nm and emission 518 nm) was measured and the concentration determined using a standard curve and expressed as μg mL$^{-1}$ of plasma.

**Flow cytometry**. For hepatic macrophage isolation, sections of liver were placed in digestion buffer with 0.25 mg mL$^{-1}$ collagenase type I, 1.2 mM CaCl$_2$, 0.1 mg mL$^{-1}$ DNase I in PBS for a minimum of 20 min, filtered through 100 μm strainer, and spun at $50 \times g$ to pellet hepatocytes. Supernatant was collected and spun at $800 \times g$ to pellet nonparenchymal cells. Cells were resuspended in histodenz (20%) and gradients generated using a $1380 \times g$ spin for 15 min with no brake. Cells at the interface were collected, washed, and stained for CD45-APC (1:50 dilution, clone 30-F11, ThermoFisher #17-0451-82), F4/80-PE (1:50 dilution, clone BM8, ThermoFisher #12-4801-82) and CD11b-FITC (1:50 dilution, clone M1/70, ThermoFisher #11-0112-41). Cells were flow sorted at the University of Colorado Cancer Center Flow Cytometry Shared Resource using the MoFlo XDP70 (Beckman Coulter, Indianapolis, IN). Resident macrophages were defined as CD45$^+$/F4/80$^{+Hi}$/ CD11b$^{+Lo}$ and recruited macrophages were defined as CD45$^+$/F4/80$^{+Lo}$/ CD11b$^{+Hi}$ [70] (gating strategy, Supplementary Fig. 4).

**BMDM isolation and stimulation**. Fresh bone marrow from the hindlimbs of each mouse was flushed with PBS, washed, and plated in macrophage media (DMEM [4.5 g L$^{-1}$ glucose, sodium pyruvate, without L-glutamine] plus 10% FBS, 2 mM L-glutamine, pen-strep, and MEM-NEAA) containing 50 ng mL$^{-1}$ M-CSF (Peprotech, Rocky Hill, NJ). Media was changed every 2–3 days. On day 7, differentiated cells (determined by adherence and morphology) were collected by gentle scraping. Cells were re-plated in macrophage media on a 24-well plate at a concentration of $1 \times 10^6$ cells mL$^{-1}$ and allowed to adhere overnight. Cells were then treated with low-dose LPS (10 ng mL$^{-1}$) for 20 h and RNA was isolated as described below.

**Bacterial phagocytosis assay**. *Listeria monocytogenes* strain 10403S expressing a GFP plasmid with Erm resistance was grown to 0.1 OD (mid-log) phase, spun at $21,000 \times g$ for 10 min at room temperature and resuspended in PBS. Adherent BMDMs were infected with bacteria at an MOI of 0.1 (vs. uninfected controls) in a 24-well plate. Cells were spun at $50 \times g$ for 5 min at 4 °C and then incubated with bacteria for 60 min at 37 °C with 5% CO$_2$. BMDMs were then washed 3X with PBS and cells were dissociated with Versene (250 μL per well; ThermoFisher) at 37 °C for 10 min with shaking at 100 rpm. Anti-CD16/32 (2.4G2 hybridoma supernatant, ATCC #HB-197) was added to block Fc receptors prior to staining in FACS buffer (1% BSA, 0.01% NaN$_3$ in PBS). All cells were stained with F4/80-PE (2 μg mL$^{-1}$, clone BM8, ThermoFisher #12-4801-82) and CD11b-APC (2 μg mL$^{-1}$, clone M1/70, ThermoFisher #17-0112-82) and then fixed in 1% paraformaldehyde for analysis on an LSRII flow cytometer (BD Biosciences, San Jose, CA). FlowJo software (BD Biosciences) was used for data analysis.

**Bacterial analysis and triglycerides in liver**. DNA was extracted from liver using a Quick-DNA/RNA kit (Zymo Research, Irvine, CA). Quantitative PCR with 16S universal bacterial primers (forward primer, 5′-TCCTACGGGAGGCAGCAGT-3′, reverse primer, 5′-GGACTACCAGGGTATCTAATCCTGTT-3′ and probe, [6-FAM]-5′-CGTATTACCGCGGCTGCTGGCAC-3′-[Iowa Black FQ]) was performed on the extracted DNA using a CFX96 instrument (Bio-Rad, Hercules, CA). PCR reactions included DyNamo ColorFlash Probe qPCR master mix (ThermoFisher), 0.2 μM of 16S universal bacterial primers and probe, and 200 ng of purified DNA. The qPCR conditions were performed with an initial denaturation of 95 °C for 10 min, followed by 40 cycles of 95 °C for 15 s and 60 °C for 90 s. For bacterial outgrowth, fresh livers were collected in 0.2% NP-40, disrupted using a tissue homogenizer (IKA Works, Wilmington, NC) for 1 min, and immediately plated on tryptic soy broth agar plates at 37 °C under normoxic conditions. Plates were incubated for 24 h and then colonies were counted and reported as the number of colony forming units. Wild-type mouse livers and diluent-only controls were used to confirm the absence of contamination. Liver triglycerides were analyzed from flash-frozen tissue. Tissue was homogenized in methanol and lipids were extracted using 1:2 methanol:chloroform followed by 0.27 volumes of distilled water, and centrifuged at maximum speed at 4 °C to separate the polar and non-polar phases. The lower, non-polar, phase was then isolated and dried under nitrogen, resuspended in isopropanol with 2% Triton X-100, directly quantified using Infinity Triglycerides Reagent (ThermoFisher) and normalized to starting tissue weight.

**RNA extraction and quantitative PCR**. RNA was isolated from frozen liver and cecum tissue or fresh BMDM cells using an E.Z.N.A. Total RNA Kit I (Omega Biotek, Norcross, GA), per instructions, except for TRIzol reagent (ThermoFisher) replaced TRK Lysis Buffer. RNA quality and quantity were determined by the Experion Automated Electrophoresis System (Bio-Rad). cDNA was synthesized from 1 μg RNA using iScript reaction mix and reverse transcriptase (Bio-Rad). The cDNA synthesis reaction was incubated at 25 °C for 5 min, 42 °C for 30 min, and 85 °C for 5 min. Quantitative PCR was performed with primer sets (Supplementary Table 2) for genes of interest and reference genes (designed with NCBI's Primer-BLAST) and iQ SYBR Supermix (Bio-Rad) using the following conditions: initial denaturation at 95 °C for 3 min, followed by 40 cycles of 95 °C for 10 s and 60 °C for 30 s. Reactions were run in duplicate on an iQ5 Real Time PCR detection

system (Bio-Rad) along with a no-template control per gene. Validation experiments were performed to demonstrate that efficiencies of target and reference genes were approximately equal. Data were normalized to *18S* rRNA and *Gapdh* using the comparative Ct method.

**SCFA and BA analyses**. Murine cecum samples were prepared for SCFA extraction as previously described[24]. Amounts of acetate, propionate, and butyrate were measured, normalized to an internal standard, and reported at mM g$^{-1}$ of cecal content. For BAs, liver and fecal samples were weighed to the nearest 0.01 mg, extracted in Optima grade methanol (ThermoFisher) at 12 mg mL$^{-1}$, agitated at 4 °C for 30 min, and centrifuged at 10,000 × g for 10 min at 4 °C. Ten microliters of supernatant was injected into a Thermo Vanquish UHPLC system and run on a Kinetex C18 column (150 × 2.1 mm, 1.7 μm) at 300 μL min$^{-1}$ and compared against isotope-labeled BA standards (50 nM each, all 2,2,4,4-deuterated, prepared in Optima methanol). Cholic acid and muricholic acid (primary BAs), taurocholic acid, glycochenodeoxycholic acid, glycocholic acid, and tauromuricholic acid (conjugated primary BAs), ursodeoxycholic acid (secondary BA), and taur-oursodeoxycholic acid (conjugated secondary BA) were measured. Solvents were A- 0.1% formic acid in water and B- 0.1% formic acid in acetonitrile, all Optima grade. The gradient was 0–1 min at 20% B, 1–2 min 20–70% B, 2–7 min 70–78% B, 7–7.1 min 78–95% B, 7.1–8 min hold at 95% B, 8–8.1 min return to 20% B, 8.1–10 min hold at 20% B. The autosampler was held at 7 °C and the column compartment at 25 °C. The UHPLC system was coupled online with a Thermo Q Exactive mass spectrometer, scanning at 70,000 resolution in the 350–550 *m/z* range in negative ion mode. Negative electrospray ionization was achieved with 4 kV spray voltage, 25 sheath gas, and 5 auxiliary gas. Data was analyzed using MAVEN[71]. Absolute quantitation was performed by measuring the ratios of labeled and endogenous BAs.

**Histological analysis and scoring**. Sections of liver were collected and fixed in 10% formaldehyde and transferred to 70% ethanol until embedded in paraffin and processed on slides for hematoxylin-eosin (H&E) staining. All staining was completed by the University of Colorado Cancer Center Research Histology Shared Resource. Images were captured on an Olympus BX53 microscope using cellSens software and DP27 camera (Olympus). For each mouse, multiple H&E stained sections of the liver were examined and one complete section from each mouse was scored. Histological analysis was quantified by a blinded pathologist using a modified Pediatric NAFLD Histological Score (PNHS), designed and validated by Alkhouri et al.[40]. Histological features were scored on a scale of 0–3 on steatosis, 0–3 on lobular inflammation, 0–2 on portal inflammation. These were combined for a total modified PNHS. Ballooning and fibrosis were not included in the scoring since they are characteristic features of NASH which was not seen in this model.

**Statistics**. Statistical analyses were conducted using Prism (GraphPad Software, La Jolla, CA). Differences between groups (NW vs. Ob) were determined by two-tailed Student's *t*-test for independent groups except for histological scoring, which was analyzed by Mann–Whitney *U*-test. Unless otherwise stated, data are expressed as mean ± SEM or as an absolute number or percentage for categorical variables, with significance set at $P < 0.05$ by Student's *t*-test.

## Data availability

16S rRNA sorted paired-end sequence data were deposited in the NCBI Sequence Read Archive under accession number PRJNA492901. All other data supporting the findings of this study are available within the paper and its supplementary files or are available from the corresponding author upon reasonable request.

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

## Acknowledgements

We appreciate the technical assistance from Erin Severs with the gnotobiotic mice and Michael Nash with assistance in tissue harvesting. We thank Sarah Farabi for assistance with diet analysis. We thank Karen Jonscher for scientific discussions and David Brumbaugh for manuscript review. This study was supported by the American Diabetes Association/Glaxo Smith Kline Targeted Research Award (1-13-GSK-13, to J.E.F., L.A B., and N.F.K.), the NIH/NCATS Colorado Clinical and Translational Sciences Institute (TL1-TR001081, to T.K.S), Children's Hospital Colorado Child Health Research Internship (to L.B.), Webb-Waring early career grant (Boettcher Foundation, to A.D.), the NIH/National Institute of Allergy and Infectious Diseases (R01AI131662, to S.E.C. and L.L.L.), and the University of Colorado GI and Liver Innate Immune Program (to D. N.F. and K.A.K.), the NIH/National Institute of Diabetes and Digestive and Kidney Diseases (R01DK07864, to L.A.B. and T.L.H.). Digestive and Kidney Diseases T32 DK007658 and NIH/Child Health and Development grant F32-0978068 (B.Y.).

## Author contributions

T.K.S., K.C.E. and J.E.F. conceived and designed the study. T.K.S., R.C.J. and J.E.F. wrote the manuscript. T.K.S. performed the experiments, unless noted otherwise. C.E.M. provided technical assistance with mice and microbiota analysis. S.E.C. designed and performed the phagocytosis and bacterial growth experiments with materials and funding from L.L.L. R.C.J. performed qPCR. L.B. provided technical assistance with mice and study design. D.I. and D.N.F. performed 16S rRNA sequencing and 16S bacterial DNA analysis. D.J.L. aided with microbiota analysis. L.K.J. provided histology imaging and scoring. T.W. performed SCFA analysis. L.A.B. and T.L.H. ran the clinical study from which samples and clinical data were taken. K.A.K. operated the GF mouse facility and assisted with experimental design. A.D. performed BA quantification. B.Y. and N.F.K. helped to design the clinical study and participated in the sample collection during the infant visits.

## Additional information

**Competing interests:** J.E.F. is a consultant to the scientific advisory board of Janssen Pharmaceuticals. All remaining authors declare no competing interests.

