## [Peer Review File · Nature Communications]

Reviewers' comments:

Reviewer #1 (Remarks to Author):

I have read this interesting manuscript that investigated the gut microbiome in infants of obese mothers increases susceptibility to obesity and NAFLD in germ-free mice.

1. Please supply the University of Colorado Institutional Animal Care and Use Committee ethics approval number.
2. Re: page 24 lines 426-428. How many mothers were prescribed antibiotics at delivery? And from which groups i.e., NW or OB mothers were treated with antibiotics for group B streptococcus infections? And which antibiotics were used? If patients were allergic to penicillin, vancomycin would have been the choice and this is a significant disrupter of the intestinal microbiome. Please clarify because as stated no antibiotics exposure during gestation and after delivery but patients treated with antibiotics at delivery, this could appear somewhat confusing to the readers.
3. Loss of bacterial DNA from fecal samples can be critical in microbiome studies, especially when conducting 16S ribosomal RNA gene (rDNA) amplicon sequencing over shotgun analysis of random DNA fragments. How old (in days) were the stool samples post collection from infants when provided to researchers at the 2-week postpartum visit?
4. Re the histology sections of the liver, how were the images captured (i.e., type of scanner)? And for each sample, how many images were captured and scored?
5. Consider commencing the discussion with what the study's main outcomes were and not more introductory remarks. This then should be followed up with the usual additional discussion points with relevant information that either supports or otherwise the outcomes reported.

Reviewer #2 (Remarks to the Author):

I have read this interesting manuscript that investigated the gut microbiome in infants of obese mothers increases susceptibility to obesity and NAFLD in germ-free mice.

1. Please supply the University of Colorado Institutional Animal Care and Use Committee ethics approval number.
2. Re: page 24 lines 426-428. How many mothers were prescribed antibiotics at delivery? And from which groups i.e., NW or OB mothers were treated with antibiotics for group B streptococcus infections? And which antibiotics were used? If patients were allergic to penicillin, vancomycin would have been the choice and this is a significant disrupter of the intestinal microbiome. Please clarify because as stated no antibiotics exposure during gestation and after delivery but patients treated with antibiotics at delivery, this could appear somewhat confusing to the readers.
3. Loss of bacterial DNA from fecal samples can be critical in microbiome studies, especially when conducting 16S ribosomal RNA gene (rDNA) amplicon sequencing over shotgun analysis of random DNA fragments. How old (in days) were the stool samples post collection from infants when provided to researchers at the 2-week postpartum visit?
4. Re the histology sections of the liver, how were the images captured (i.e., type of scanner)? And for each sample, how many images were captured and scored?
5. Consider commencing the discussion with what the study's main outcomes were and not more introductory remarks. This then should be followed up with the usual additional discussion points with relevant information that either supports or otherwise the outcomes reported.

Reviewer #3 (Remarks to the Author):

Summary:

The authors have studied the association between increased risk for offspring obesity and subsequent non-alcoholic fatty liver disease (NAFLD). They compared germ-free mice colonized with stool microbes from 2-week-old infants born to obese (Inf-ObMB) or normal-weight (Inf-NWMB) mothers and found that Inf-ObMB-colonized mice showed increased hepatic mRNA expression for genes in endoplasmic reticulum stress and innate immunity together with histological signs of periportal inflammation. Also, Inf-ObMB mice showed increased intestinal permeability, reduced macrophage phagocytosis, and dampened cytokine production suggestive of a dysfunctional macrophage compartment. However, liver macrophages showed a pro-inflammatory phenotype whereas BMDM show hypo-inflammatory status. Exposure to a Western-style diet in Inf-ObMB mice promoted excess weight gain, steatosis, and accelerated NAFLD. Hence, dysbiotic microbiome in infants born to obese mothers causes increased susceptibility to obesity and NAFLD associated with a dysfunctional macrophage phenotype. They claim that dysbiotic microbiome is a possible initiating factor to childhood obesity and NAFLD. The main finding of this study is that the altered gut microbiota in 2-week-old infants born to Obese mothers induced metabolic and inflammatory changes in the liver and bone marrow cells of mice colonized with these dysbiotic microbes. Unfortunately, this finding is descriptive and lack mechanistic immunological insights on how this effect is mediated. The beauty of this study is that the mouse data phenocopy clinical NAFLD, thereby providing an animal model to further elucidate the molecular mechanisms of NAFLD and obesity development.

Major concerns:

- The authors describe altered gene expression and inflammatory mediators in mice colonized with stool from 2-week-old infants born to Obese mothers. Moreover, the authors studied liver and BM macrophages and showed alterations in number and function. However, the role of the inflammatory mediators and the contribution of macrophages for the inflammatory response and the disease development remains elusive. Hence, the study describes many phenomena, but mechanistic insights on how the dysbiosis mediate inflammation and disease has not been studied.
- The authors found increased liver inflammation in ObMB mice (Figure 4) and an altered macrophage compartment. However, the causal connection between macrophages and inflammation remains to be studied. Are liver macrophage numbers beneficial or detrimental to hepatocytes in this model? Cell sorting of tissue and recruited macrophages and performing functional analysis of these cells (transcriptomic or proteomic level) would provide further insights into the role of these cells. Moreover, specific targeting of these molecules and / or deletion of tissue macrophages (anti-CSF1R antibody intravenously) or recruited macrophages (e.g. low dose of clodronate liposomes or mice deficient for CCR2) would address this question.
- BMDMs from ObMB mice produced decreased amounts of proinflammatory cytokines and are less phagocytic. Studying BMDMs seems to be far-fetched. How do these cells contribute to the disease development? Why have the authors decided to study these cells?
- Data in Figure 5C are not convincing. The uninfected Inf-NWMB control group shows a background fluorescence of 17.3% as opposed to the Inf-ObMB mice (0.3%). This discrepancy needs to be explained. Moreover, percentage-wise the difference between the uninfected and infected Inf-NWMB mice is much stronger compared to Inf-ObMB mice indicating increased infection in NM mice. The authors claim that „BMDMs from Inf-ObMB-colonized mice were significantly less efficient in phagocytosing *Listeria* compared with BMDMs from Inf-NWMB mice (P=0.04; Fig. 5B, C), consistent with reduced bacteria clearance and inflammation in Inf-ObMB mice.“ Hence, this conclusion is not supported by the data.
- Western diet in Inf-ObMB mice does not change the overall number of macrophages. However, the number of tissue MP numbers were increased, whereas the number of recruited MPs were decreased. Also, the levels of IL-1b and TNF were increased. As mentioned in the beginning, these data are descriptive and a causal connection is missing. Are these mediators produced by tissue or

by recruited macrophages? And how do these mediators contribute to the disease development? The experimental approaches that tackle these questions are specific targeting of the molecules (e.g. CX3CR1 cre x Targeting flox mice).

- The authors show that there is a trend towards more bacterial growth in livers from Inf-ObMB-colonized mice. These data are based on the colonies that have grown on tryptic soy broth agar plates. I suggest to perform 16S qPCR to corroborate the notion of bacterial translocation from the gut.
- The authors claim that cecal SCFA concentrations influence the phenotype of BMDMs. How are these molecules transported into the BM? Is it possible to measure blood SCFA levels? If so, it would be interesting to incubate BMDMs with physiological SCFA levels to show if they are responsible for BMDM desensitization or are some translocated bacteria and/or PAMPs responsible for the effect.
- After exposure to WSD, there were no more differences in gut microbiota between Inf-ObMB-colonized and Inf-NWMB-colonized mice after 6 weeks, still the Inf-ObMB-colonized mice had higher bodyfat level and higher liver triglyceride level suggestive of the role of early immunological priming on the described phenotype. I am missing in the discussion which cell could be most affected by this early immunological priming and also what would be the connection/ contribution of gut resident immune cells to the liver metabolism and liver resident macrophages. Another question addresses the development of the mice (weight gain; liver metabolism) at later time points than 6 weeks. Does the initial priming promote further disease development?

August 7, 2018

MS: NCOMMS-18-11530A

TITLE: The gut microbiome in infants of obese mothers increases susceptibility to obesity and NAFLD in germ-free mice

Summary of changes: We thank the reviewers and the Editor for their helpful suggestions and the opportunity to re-submit our manuscript. We have reviewed each comment in detail and made the critical changes suggested by each reviewer. We also gratefully acknowledge the Editor's comments that no further experimental work is required for this communication, but that we address each reviewer's concerns which we have done below. In response to Reviewer 1, we have added details of the subjects and histological analysis. In reply to Reviewer 2's valuable suggestions for mechanistic studies to address the role of recruited macrophages in the phenotype of Inf-ObMB mice, we completely agree that capturing the recruited or tissue macrophages and their transcriptomic/proteomic signature would be highly informative; however, we don't have these cells available. We did however identify the source of critical pro-inflammatory cytokines IL1b and TNFa in recruited macrophages in Inf-ObMB-WSD mice (Fig. 6B) and include possible alternative mechanisms for our findings in the revised discussion. As suggested by Reviewer 2, we have added new data on 16S rRNA quantification using the liver DNA to quantify bacteria (Fig. 2F). We found increased bacteria in the liver of the In-ObMB mice that further supports our finding that a leaky gut in the Inf-ObMB mice contributes to the translocation of microbes and inflammation in the liver. In response to Reviewer 3 we have provided 2 new histological images and utilized a modified pediatric NAFLD Histological Score (PNHS) and appropriate statistical test to show differences in each category based on histological scoring criteria. We also dampened the tone of our conclusions regarding periportal inflammation as indicative of pediatric NAFLD. A complete list of revisions, including a point by point response to the reviewers' comments is below. We would, of course, be willing to make further revisions based on the Editor or reviewers' suggestions to make our manuscript suitable for publication in *Nature Communications*.

Response to Individual Reviewers:

Referee 1 (Comments to Authors):

I have read this interesting manuscript that investigated the gut microbiome in infants of obese mothers increases susceptibility to obesity and NAFLD in germ-free mice.

We thank the reviewer for the effort and time in reviewing our manuscript and we are delighted it was found of interest.

1. Please supply the University of Colorado Institutional Animal Care and Use Committee ethics approval number.

The studies were approved by the University of Colorado Institutional Animal Care and Use Committee (IACUC) protocol number 00309 added on page 25 Line 451.

2. Re: page 24 lines 426-428. How many mothers were prescribed antibiotics at delivery? And from which groups i.e., NW or OB mothers were treated with antibiotics for group B streptococcus infections? And which antibiotics were used? If patients were allergic to penicillin, vancomycin would have been the choice and this is a significant disrupter of the intestinal microbiome. Please clarify because as stated no antibiotics exposure during gestation and after delivery but patients treated with antibiotics at delivery, this could appear somewhat confusing to the readers.

One normal-weight mother received antibiotic treatment during delivery – 200 mL of 5 million units Penicillin G potassium in normal saline approximately 3 hours prior to delivery. One woman in the obese group received a single dose of 1,000 mg cefazolin IV in the immediate postpartum period. This information was added to the manuscript on page 25 line 460.

3. Loss of bacterial DNA from fecal samples can be critical in microbiome studies, especially when conducting 16S ribosomal RNA gene (rDNA) amplicon sequencing over shotgun analysis of random DNA fragments. How old (in days) were the stool samples post collection from infants when provided to researchers at the 2-week postpartum visit?

We have included a brief summary of stool sample collection in the manuscript. Infant stool was collected within a 24-hour window prior to the 2-week postpartum visit - this was added on page 26 line 477 and we added the reference to our previous publication that describes the collection in depth (“Infant stool samples were collected 24 h before the 2-wk visit by mothers with the use of a nonabsorbent liner placed in the infant’s diaper. When the diaper contained stool, mothers removed the liner, placed it in a plastic bag, and stored it at -20°C until the study visit” from PMID: 27140533).

4. Re the histology sections of the liver, how were the images captured (i.e., type of scanner)? And for each sample, how many images were captured and scored?

The images were captured on an Olympus microscope (BX53) using cellSens software, also by Olympus. The Olympus camera that created the photos was a DP27. For each mouse, we examined multiple H&E stained sections of the liver and one complete section from each mouse was scored. This information has been added to the Methods section on page 30 line 586.

5. Consider commencing the discussion with what the study’s main outcomes were and not more introductory remarks. This then should be followed up with the usual additional discussion points with relevant information that either supports or otherwise the outcomes reported.

We thank the reviewer for this suggestion. We have modified the first paragraph of the discussion as follows (page 20 line 320): “The main finding of this study is that the altered gut microbiota in 2-week-old infants born to Ob mothers induced metabolic and inflammatory changes in the liver and bone marrow cells of GF mice colonized with these dysbiotic microbes. Importantly, these mice were predisposed to accelerated weight gain and development of fatty liver following exposure to a WSD. Clinical data support correlations between pediatric obesity, NAFLD, and gut dysbiosis; however, this is the first experimental evidence in support of the hypothesis that changes in the gut microbiome in infants born to obese mothers precede or even directly initiate and promote these disease pathways.”

Referee 2 (Comments to authors):

Summary:

The authors have studied the association between increased risk for offspring obesity and subsequent non-alcoholic fatty liver disease (NAFLD). They compared germ-free mice colonized with stool microbes from 2-week-old infants born to obese (Inf-ObMB) or normal-weight (Inf-NWMB) mothers and found that Inf-ObMB-colonized mice showed increased hepatic mRNA expression for genes in endoplasmic reticulum stress and innate immunity together with histological signs of periportal inflammation. Also, Inf-ObMB mice showed increased intestinal permeability, reduced macrophage phagocytosis, and dampened cytokine production suggestive of a dysfunctional macrophage compartment. However, liver macrophages showed an pro-

inflammatory phenotype whereas BMDM show hypo-inflammatory status. Exposure to a Western-style diet in Inf-ObMB mice promoted excess weight gain, steatosis, and accelerated NAFLD. Hence, dysbiotic microbiome in infants born to obese mothers causes increased susceptibility to obesity and NAFLD associated with a dysfunctional macrophage phenotype. They claim that dysbiotic microbiome is a possible initiating factor to childhood obesity and NAFLD.

The main finding of this study is that the altered gut microbiota in 2-week-old infants born to Obese mothers induced metabolic and inflammatory changes in the liver and bone marrow cells of mice colonized with these dysbiotic microbes. Unfortunately, this finding is descriptive and lack mechanistic immunological insights on how this effect is mediated. The beauty of this study is that the mouse data phenocopy clinical NAFLD, thereby providing an animal model to further elucidate the molecular mechanisms of NAFLD and obesity development.

We thank the reviewer for the thoughtful assessment of our manuscript. We have addressed the concerns for potential mechanisms below.

Major concerns:

- The authors describe altered gene expression and inflammatory mediators in mice colonized with stool from 2-week-old infants born to Obese mothers. Moreover, the authors studied liver and BM macrophages and showed alterations in number and function. However, the role of the inflammatory mediators and the contribution of macrophages for the inflammatory response and the disease development remains elusive. Hence, the study describes many phenomena, but mechanistic insights on how the dysbiosis mediate inflammation and disease has not been studied.

We thank the reviewer for bringing up the potential mechanisms for the inflammatory response. Regarding the mechanisms for how infant dysbiosis mediates inflammation and the role of macrophages, our data demonstrate that there are many potential molecular pathways or factors to consider. We identified gut bacteria translocation, changes in bile acids and short-chain fatty acids, and hematopoietic stem cell programming as candidate factors that might contribute to Inf-ObMB mouse liver inflammation and progression to NAFLD. We are pursuing each one of these mechanisms individually. Identifying the offending inflammatory mediator(s) and the population of macrophages involved will require quite a bit more work that we hope to publish in a subsequent follow-up publication. We have addressed some of these possibilities throughout the revised discussion, largely on page 23 line 394.

- The authors found increased liver inflammation in ObMB mice (Figure 4) and an altered macrophage compartment. However, the causal connection between macrophages and inflammation remains to be studied. Are liver macrophage numbers beneficial or detrimental to hepatocytes in this model? Cell sorting of tissue and recruited macrophages and performing functional analysis of these cells (transcriptomic or proteomic level) would provide further insights into the role of these cells. Moreover, specific targeting of these molecules and / or deletion of tissue macrophages (anti-CSF1R antibody intravenously) or recruited macrophages (e.g. low dose of clodronate liposomes or mice deficient for CCR2) would address this question.

We thank the reviewer for these helpful suggestions for further experiments to address the possible role of resident vs. recruited macrophages as potential sources of hepatic inflammation in Inf-ObMB mice. We did isolate the recruited macrophages in the Inf-ObMB mice on WSD (Fig. 6E) and found increased proinflammatory cytokine expression levels of *Tnf* and *Il1b* compared with Inf-NWMB mice, suggesting that the recruited macrophages in Inf-ObMB liver on WSD actively participate in liver inflammation. However, we agree with the reviewer about characterizing the cell's transcriptomic/proteomic signature in the Inf-ObMB mice as the recruited macrophage population could be phenotypically and functionally distinct from the resident Kupffer

cells and either cell type might contribute to the pathogenesis of the increased liver inflammation, and we have noted this in our discussion on page 21 line 365. Unfortunately, we no longer have these cells available.

We could repeat our experiments and deplete these cells as suggested, which has been shown to suppress hepatic steatosis and inflammation in mouse models of NAFLD (PMID: 28940700; PMID: 26908374), but another strategy for addressing how infant dysbiosis-induced inflammation increases NAFLD is to focus on the differences in microbes in germ-free mice, which we are currently pursuing using our model. Given that bacteria escaping the gut and reaching the liver contributes to liver inflammation, as has been shown in NAFLD, independent of obesity (PMID: 29362454; PMID: 29956209), we aim to address whether the differences in Proteobacteria vs. Clostridia alter the gut and liver macrophage populations to help answer the reviewer's question. These experiments are underway, and we plan to publish those studies in a follow-up publication. We have highlighted the importance of Proteobacteria and Clostridia as potential drivers of this phenotype in our discussion on page 23 line 416.

- BMDMs from ObMB mice produced decreased amounts of proinflammatory cytokines and are less phagocytic. Studying BMDMs seems to be far-fetched. How do these cells contribute to the disease development? Why have the authors decided to study these cells?

Bone marrow hematopoiesis is the definitive source of circulating monocytes and in the liver, F4/80 low macrophages are continuously replaced by bone marrow-derived progenitors (PMID: 23619691; PMID: 29117177). Circulating monocytes derived from the bone marrow are recruited to a target tissue throughout the duration and resolution of an inflammatory response, and therefore, there is substantial interest in delineating the specific tasks for bone marrow-derived macrophages in inflammation, phagocytosis, and bacterial clearance in NAFLD. In studying macrophages derived from the bone marrow, we see the cytokine response and phagocytosis persisting in cell culture 7 days after isolation from the animal. This illuminates the differences that are programmed at the level of the bone, rather than changes resulting from the systemic environment either during circulation as a monocyte or in the liver as a recruited macrophage.

We used bone marrow-derived macrophages as they provide an *in vitro* system where we can produce larger quantities of cells to study than can be isolated from the mouse liver and provide new immunological insights into the programming due to the dysbiotic microbiome rather than the response of the cell of the host environment. Studies by Khosravi et al. (PMID: 24629343) demonstrated the influence of the gut microbiome on bone marrow hematopoiesis and its ability to control bacterial infection. Luo et al. (PMID: 26387866) demonstrated the ability of a high-fat diet altered microbiota to impact hematopoiesis in the bone marrow niche, increasing its adiposity. These studies together indicate a need to study the role of the microbiome on the bone marrow-derived immune cells, specifically precursors to the recruited macrophages, which have a demonstrated, critical role in NAFLD development. We have clarified our rationale and approach in the revised results section on page 15 line 251.

- Data in Figure 5C are not convincing. The uninfected Inf-NWMB control group shows a background fluorescence of 17.3% as opposed to the Inf-ObMB mice (0.3%). This discrepancy needs to be explained. Moreover, percentage-wise the difference between the uninfected and infected Inf-NWMB mice is much stronger compared to Inf-ObMB mice indicating increased infection in NM mice. The authors claim that „BMDMs from Inf-ObMB-colonized mice were significantly less efficient in phagocytosing *Listeria* compared with BMDMs from Inf-NWMB mice (P=0.04; Fig. 5B, C), consistent with reduced bacteria clearance and inflammation in Inf-ObMB mice.“ Hence, this conclusion is not supported by the data.

We apologize, this figure was mislabelled. The corrected version of Figure 5 has been uploaded, and the quantification in Fig. 5B is correct. Hence, our conclusion here still stands. Both the Inf-NWMB and Inf-ObMB have limited CD11b, Lm-GFP positive cells in the uninfected controls (0.952% and 0.353%, respectively) whereas

the Inf-NWMB bone marrow-derived macrophages have 22.5% uptake of CD11b, Lm-GFP positive cells compared to the significantly reduced 17.3% in the Inf-ObMB bone marrow-derived macrophages.

- Western diet in Inf-ObMB mice does not change the overall number of macrophages. However, the number of tissue MP numbers were increased, whereas the number of recruited MPs were decreased. Also, the levels of IL-1b and TNF were increased. As mentioned in the beginning, these data are descriptive and a causal connection is missing. Are these mediators produced by tissue or by recruited macrophages? And how do these mediators contribute to the disease development? The experimental approaches that tackle these questions are specific targeting of the molecules (e.g. CX3CR1 cre x Targeting flox mice).

We thank the reviewer for these helpful suggestions and point out that in Fig. 6B, we measured increased expression of *I11b* and *Tnf*, **in the recruited macrophages in the Inf-ObMB mice**. In addition to their role in inflammation, these cytokines might also contribute to NASH pathogenesis by promoting insulin resistance and altering lipid metabolism (PMID: 18929493; PMID: 20347818). As the reviewer notes, more sophisticated future studies will be required to determine the degree to which interfering with recruitment of CCR2+ recruited macrophages contribute to microbial-induced changes in NAFLD.

- The authors show that there is a trend towards more bacterial growth in livers from Inf-ObMB-colonized mice. These data are based on the colonies that have grown on tryptic soy broth agar plates. I suggest to perform 16S qPCR to corroborate the notion of bacterial translocation from the gut.

Both reviewer 2 and 3 suggested we perform 16S qPCR on the liver to assess total bacterial quantity, and we have done so. We found significantly more bacterial DNA in the livers of the Inf-ObMB mice and have made this Fig. 2F since it is more inclusive than the data from the outgrowth of bacteria from the livers under one growth condition. We moved the bacterial growth data to Supplementary Fig. 1A.

Added to page 10 line 172: We quantified total bacteria in Inf-NWMB and Inf-ObMB mouse livers through measurement of total 16S DNA, as 16S is only found in bacterial DNA. Significantly more 16S DNA was measured in Inf-ObMB livers compared with Inf-NWMB ($P < 0.0001$, Fig. 2F). Added to page 28 line 543: DNA was extracted from liver using a Quick-DNA/RNA Kit (Zymo Research, Irvine, CA) and quantitative PCR with 16S total bacterial primers was utilized as previously (PMID: 27199534) to detect and quantify bacterial DNA.

- The authors claim that cecal SCFA concentrations influence the phenotype of BMDMs. How are these molecules transported into the BM? Is it possible to measure blood SCFA levels? If so, it would be interesting to incubate BMDMs with physiological SCFA levels to show if they are responsible for BMDM desensitization or are some translocated bacteria and/or PAMPs responsible for the effect.

The reviewer brings up an important point about the causes for bone marrow-derived macrophage desensitization. Although we did not measure butyrate in circulation, it is tempting to speculate, based on our data showing increased butyrate in the intestine, reduction in gut barrier activity, and bone marrow-derived macrophage desensitization, that these events are linked, given butyrate's well-known effect on dampening macrophage responsiveness to LPS (PMID: 24434557). However, other explanations for how the microbiota affect hematopoiesis might include microbial-associated or pathogen-associated molecular patterns (MAMPs or PAMPs) or other growth factors (PMID: 24629343). Although butyrate has been clearly established in published studies described thus far, the studies are based on *ex vivo* analysis of bone marrow or peripheral populations, which might not necessarily fully recapitulate the physiologic conditions of a live organism. We have acknowledged this in the revised discussion (page 21 line 352). Further work is needed to uncover the complex molecular mechanisms by which the products of the bacteria, be it butyrate or some other signal from the gut,

impact the hematopoietic system. We have addressed a variety of possible alternative mechanisms in the revised discussion on page 23 line 394.

- After exposure to WSD, there were no more differences in gut microbiota between Inf-ObMB-colonized and Inf-NWMB-colonized mice after 6 weeks, still the Inf-ObMB-colonized mice had higher bodyfat level and higher liver triglyceride level suggestive of the role of early immunological priming on the described phenotype. I am missing in the discussion which cell could be most affected by this early immunological priming and also what would be the connection/ contribution of gut resident immune cells to the liver metabolism and liver resident macrophages. Another question addresses the development of the mice (weight gain; liver metabolism) at later time points than 6 weeks. Does the initial priming promote further disease development?

The reviewer raises an important question about how gut “immunological priming” by early differences in the microbiome might be affecting liver metabolism. The influence of gut microbiota on hepatic energy metabolism is multifactorial. Different targets involving immunity (PMID: 26464517), bioactive lipid production (PMID: 29942096; PMID: 25764541), or bacterial metabolites (PMID: 26600078) have been highlighted in recent studies. Rodent studies have demonstrated a potential early causative role for an impaired intestinal immune system in the development of metabolic disease (PMID: 17456850; PMID: 27617200). The innate immune system is the first and fastest line of adaptation to changes in the microbiome, followed by the adaptive immune system, which we did not investigate in this study. We have acknowledged that other immune cell types are likely influenced by early dysbiosis on page 22 line 369. It is not possible to address them all here. The “fire” of this early inflammatory state has “gasoline” thrown onto it when the Inf-ObMB animals are exposed to a WSD. Continued consumption of lipid, simple sugar, and cholesterol beyond 6 weeks may accelerate the already inflamed liver to steatosis and pro-fibrotic state, while the Inf-NWMB mice were relatively protected at 6 weeks of exposure. We have addressed these points in the revised discussion beginning on page 23, line 403.

Referee 3 (Comments to Authors):

This is an interesting paper that strives to provide experimental evidence for the often-reported supposition that early life gut microbial community composition influences susceptibility to diet-associated steatohepatitis. It is of value to the field but has to be careful not to overstate the translatability of the mouse model, particularly the histological findings. There are also corrections needed in the description of histological scoring and in the statistical analysis of the histology. Specific comments are below.

We thank the reviewer for the thoughtful and detailed review of our histological results. We have gone through the manuscript and toned-down language where the conclusions were stated too strongly, or inappropriate conclusions were claimed. We have made major revisions to our histology figures, the details of which are discussed below.

Introduction

P5, Line 96 “causal”

We thank the reviewer for catching this spelling error – it has been fixed (page 5 line 96).

Results

p10 Lines 173-178

What is the significance of “greater variation in four types of bacteria” in liver cultures from ObMB vs NWMB-colonized mice? What types of bacteria? Is greater variation “bad” (isn’t diversity “good” in the context of the gut?) The two representative images in Supplemental Fig 1 do not add to the interpretation of this data, as a

selected image can be representative (illustration purposes) but is not convincing of a true difference between groups- need some numbers here, especially since no statistical difference in CFU/ml was found (line 75, Fig2F).

We agree with this insight and have done 16S qPCR on the liver to assess total bacteria quantity. We found significantly more bacterial DNA in the livers of the Inf-ObMB mice and have made this Fig. 2F since it is more inclusive than the data from the outgrowth of bacteria from the livers under one growth condition (now Supplementary Fig. 1A). These new data are now included in the revised manuscript on page 10 line 172 and Fig. 2F.

P13 Lines 227-228 “protein deposition around the bile ducts”

? Where is the evidence of this and what does that even mean? The image of Inf-ObMB in Fig 2E shows a mildly dilated portal vein with serous contents (normal appearance post-euthanasia), some mildly dilated peribiliary lymphatics (also within normal limits post-euthanasia), and minimal lymphocytic periportal, peribiliary infiltration (see comment below). Please explain and provide evidence of the protein deposition claim. Did these mice have steatosis – this is not discussed.

We have removed our claim of protein deposition and have focused on the periportal inflammation, as outlined below. There was no evidence for micro or macro steatosis either biochemically (triglyceride assay) or histologically in the Inf-ObMB mice compared to Inf-NWMB mice. This is to be expected given the mice are on normal chow and only exposed to the microbes for 21 days at the time of analysis. On Page 13 line 235, we added the following “No micro- or macro-steatosis was present in either group, as expected given chow diet consumption and the short 21-day colonization period.”

P13 Lines 229-230 “this histologic finding (periportal inflammation) is unique to the livers of children with NAFLD”: While lymphocytic periportal/peribiliary infiltration may be unique to pediatric NAFLD in humans, it is actually a fairly common leukocyte infiltration pattern in mice (Thoolen et al, 2010 Tox Path PMID: 21191096) that can be seen idiopathically and at increased frequency and severity with certain pathogens (ex. *Helicobacter hepaticus*), with age, or non-specifically with gut dysbiosis of many causes. Periportal/peribiliary infiltration related to gut bacterial transfer may very well be present in this study, since in mice this finding is often attributed to bacterial translocation from the gut to the liver via the portal circulation. Given the frequency of this finding in mice, though, I am leery of statements implying that it is diagnostic (or “unique”) to NAFLD in this species. In your discussion you state that this finding is associated with advanced pediatric NAFLD. Fig 2E does not look like advanced NAFLD- did these mice have steatosis? Additionally, at 8-10 weeks of age, these are adult mice, albeit young adult mice. Can you explain why you would be more likely to see a pediatric (periportal) pattern of NAFLD-related inflammation rather than an adult (lobular) pattern in adult mice? I suggest making the language more circumspect about the significance of this finding, in terms of confirming an NAFLD of pre-NAFLD-like state in mice.

We thank the reviewer for this in-depth and well researched explanation. We agree with the reviewer that the animals did not show advanced NAFLD and have changed the language about how the initial periportal inflammation in our study might relate to the pathogenesis of NAFLD to the following (Page 20 line 332). “This periportal vein inflammation is seen in rodents with bacterial translocation and in humans with certain forms of pediatric NAFLD and features of the metabolic syndrome, suggesting that it might be clinically relevant as an early manifestation of leaky gut”. Given that our findings in mice are derived from stool of infants at 2 weeks of age, our results suggest this early inflammation might, if present in these infants born to obese mothers, initiate the pathogenic mechanisms associated with NAFLD following a secondary hit, as suggested by the accelerated inflammation and steatosis in Inf-ObMB mice on 6 weeks of WSD. Notably, these differences, both biochemically and histologically, were made in adult GF mice, and might differ if we used younger mice.

P14 & p 18 Figure 4 and Figure 6. Vascular profiles pictured in Fig 4E, in 6F InfNWMB (only the vascular profile to the left), and 6F InfObMB (only the vascular profile at the bottom) are labeled as portal veins but lack accompanying hepatic arterioles and bile ductules. These vascular structures appear to be central veins. Please correct. Figure 4E should be replaced with an image that shows a comparable portion of the liver to 4F (ie. portal area) for better comparison to the region with periportal/peribiliary infiltrates show in 4F.

Thank you for this suggestion. We have ensured that the central and portal veins are labelled accurately in Fig. 6F. We have replaced both images in Fig. 4E to show comparable regions of the liver.

P18 Figure 6G for the HFD mice, although difficult to see at this magnification, shows periportal/peribiliary leukocyte infiltrates but also shows a midzonal infiltrate (black arrow, top center) more consistent with lobular inflammation of “classical” NAFLD. How does this relate to your observations of periportal distribution from the first bacterial transfer experiment? What was the predominant inflammatory pattern in the WSD-fed InfObMB mice? Do these mice have a mixed lobular and periportal pattern (also reported in pediatric NAFLD- PMID 26638195)

We thank the reviewer for bringing this important additional histological observation to our attention – we were focused on the periportal inflammation but recognize the extended pattern of lobular inflammation in the Inf-ObMB mice following a WSD. We have re-scored both areas of inflammation and have changed our description to include the following in the results (page 17 line 285): “Following 6 weeks of WSD, histologically, livers from Inf-ObMB mice demonstrated a mixed pattern of periportal and lobular inflammation and steatosis (Fig. 6F) which resulted in a near doubling of the modified PNHS compared with Inf-NWMB WSD mice ($P=0.09$; Fig. 6G; Supplementary Fig. 3B), consistent with increased hepatic cytokine expression results (Fig. 6E) and triglyceride levels (Fig. 6C).”

P18 Fig 6G It would be enlightening to see the composite scores (steatosis, inflammation, ballooning degeneration) making up the NAS, perhaps as a supplemental file. Was the major finding steatosis? Was there any fibrosis in these mice (was this scored?).

We thank the reviewer for this suggestion. We have added the composite scoring as a supplementary figure to address this (Supplementary Fig. 3B). We used a modified Pediatric NAFLD Histological Score (PNHS), designed and validated by Alkhoury et al. (PMID: 22871498). The scoring showed that all three phases of the modified scoring (discussed below) contributed to an increased PNHS.

Fig 6G and Methods (Statistics) Histological scoring by ordinal ranking requires non-parametric comparison. Although numerical scores are generated, ordinal ranks are not a continuous measure (you are not physically counting or measuring something) but are an ordered version (mild, moderate, severe) of categorical scoring. Non-parametric measures (Mann-Whitney U) should be used instead of t test. (PMID 23558974)

We thank the reviewer for bringing this to our attention. We have re-scored our histology utilizing a modified version of the published scoring system, Pediatric NAFLD Histological Score (PNHS) (PMID: 22871498). The following changes have been made to the results, methods, figure legends and figures.

Results page 17 line 287: We modified the sentence to read modified PNHS.

Methods page 30 line 588: We changed the wording to say “Histological analysis was quantified by a blinded pathologist using a modified Pediatric NAFLD Histological Score (PNHS), designed and validated by Alkhoury et al. (PMID: 22871498). In brief, histological features were scored on a scale of 0-3 on steatosis, 0-3 on lobular inflammation, 0-2 on portal inflammation. These were combined for a total modified PNHS. Ballooning and fibrosis were not included in scoring since they are characteristic features of NASH which was not seen in this model.”

Methods page 31 line 598: We added “Histology scoring was analysed by Mann-Whitney U test” to the Statistics section.

Fig. 6 legend page 18 line 309: We have edited the figure legend for Figure 6, adding in that G was analysed using two-tailed Mann-Whitney U test.

The y axis on Fig. 6G has been relabelled “modified PNHS”.

Discussion

P20 Line 316 “predisposed to fatty liver” It would be helpful to show the steatosis comparison (the subscores for steatosis contributing to the NAS) to establish this.

Please see above. Although the steatosis score was not significant by modified PNHS, liver triglycerides were significantly elevated.

P20 Lines 321-323 “...periportal inflammation is a unique characteristic that may be clinically relevant as often found in pediatric NAFLD...” As discussed above, periportal/peribiliary infiltration is a common finding in mice (PMID: 21191096 see Fig 81 and related text) and is by no means restricted to livers with steatosis. Your images from 6F show lesions more characteristic (than 2E) of NAFLD, but the inflammatory pattern also appears more mixed here – with both lobular and periportal infiltrates. I suggest you tone down the implications that periportal/peribiliary inflammation is somehow indicative of NAFLD in mouse models (since the normal chow animals did not have steatosis, at least from the image shown) and indicate simply that this infiltrate, while not unique to NAFLD in mouse models, may reflect dysbiosis, which may precede NAFLD with environmental stimulus (diet).

We are in agreement with this statement and this change has been made and described above.

Methods

P 25 Lines 447-450 Is there a reason you picked C57BL/6J mice rather than an NNT-competent B6 strain? There is some evidence that, although they gain weight on a high fat diet, B6J mice have opposite response to some aspects of diet-associated metabolic stress than would be expected with insulin resistance and HFD consumption, particularly on diets with relatively moderate fat content (PMID 27495226; 20057372)

It is very clear that the strain matters; however, we are limited by the availability of mice in the germ-free facility. At this point only C57BL/6J mice have been derived into germ-free mice in our core.

P29 Lines 542-549 Histology

Histologic evaluation is described as scoring steatosis, inflammation, and fibrosis on a 0-3 scale and generating a summarized NAFLD activity score. NAFLD activity score (NAS) is a standardized parameter calculated from a well-defined scoring system described by Kleiner (PMID 15915461), which summarizes scores for steatosis (0-3), inflammation (0-3), and ballooning degeneration (0-2) to generate NAS (range 0-8). This system scores fibrosis

separately (not part of NAS). Please provide a reference for the scoring system used. If it was developed specific for this project, please describe the specific criteria used to separate the different scores and remove the summary descriptor “NAFLD activity score”, as this is typically associated specifically with Kleiner scoring. If Kleiner scoring (NAFLD activity score) was used please correct the description of the scoring points distribution (fibrosis is not part of NAS in this system).

We removed the NAS references throughout the manuscript and have re-scored our histology using a validated and published scale, as described above.

P29 Lines 551-555 Statistics

See comment above under Fig 6G. A non-parametric test is needed for comparison of the histological scores (ordinal scores are not continuous measures).

This change has been made and described above.

REVIEWERS' COMMENTS:

Reviewer #2 (Remarks to the Author):

I have read the comments from all the reviewers and the rebuttals as well as the revised version of the manuscript and as such I have no further queries for the authors.

Reviewer #3 (Remarks to the Author):

Dear Authors,

thank you for considering my suggestions. I have no further comments.

Best Regards

Reviewer #4 (Remarks to the Author):

I am satisfied with the responses to previous comments and feel the report is now suitable for publication.